# Mechanistic basis of the inhibition of SLC11/NRAMP-mediated metal ion transport by bis-isothiourea substituted compounds

Cristina Manatschal[1†*], Jonai Pujol-Giménez[2,3,4†], Marion Poirier[5†], Jean-Louis Reymond[5], Matthias A Hediger[2,3,4], Raimund Dutzler[1*]

[1]Department of Biochemistry, University of Zurich, Zurich, Switzerland; [2]Institute of Biochemistry and Molecular Medicine, University of Bern, Bern, Switzerland; [3]Membrane Transport Discovery Lab, Department of Nephrology and Hypertension, Inselspital, University of Bern, Bern, Switzerland; [4]Department of Biomedical Research, University of Bern, Bern, Switzerland; [5]Department of Chemistry and Biochemistry, University of Bern, Bern, Switzerland

*For correspondence:
c.manatschal@bioc.uzh.ch (CM);
dutzler@bioc.uzh.ch (RD)

†These authors contributed equally to this work

Competing interests: The authors declare that no competing interests exist.

**Abstract** In humans, the divalent metal ion transporter-1 (DMT1) mediates the transport of ferrous iron across the apical membrane of enterocytes. Hence, its inhibition could be beneficial for the treatment of iron overload disorders. Here we characterize the interaction of aromatic bis-isothiourea-substituted compounds with human DMT1 and its prokaryotic homologue EcoDMT. Both transporters are inhibited by a common competitive mechanism with potencies in the low micromolar range. The crystal structure of EcoDMT in complex with a brominated derivative defines the binding of the inhibitor to an extracellular pocket of the transporter in direct contact with residues of the metal ion coordination site, thereby interfering with substrate loading and locking the transporter in its outward-facing state. Mutagenesis and structure-activity relationships further support the observed interaction mode and reveal species-dependent differences between pro- and eukaryotic transporters. Together, our data provide the first detailed mechanistic insight into the pharmacology of SLC11/NRAMP transporters.

## Introduction

Hereditary hemochromatosis (HH) is a multigenic iron overload disorder that results from the excessive absorption of iron in the intestine (*Pietrangelo, 2010*; *Yen et al., 2006*). In the absence of a regulated mechanism for its excretion, excessive iron can lead to significant tissue damage in the heart, liver, endocrine glands and other organs (*Pietrangelo, 2010*; *Yen et al., 2006*). The most prevalent form of HH is associated with the upregulation of the iron transport protein DMT1 (or SLC11A2) (*Byrnes et al., 2002*; *Fleming et al., 1999*; *Rolfs et al., 2002*; *Stuart, 2003*), which facilitates the uptake of ferrous iron ($Fe^{2+}$) across the apical membrane of enterocytes and whose expression is regulated on a transcriptional level (*Fleming et al., 1997*; *Gunshin et al., 1997*; *Shawki et al., 2012*). The current strategy to treat hemochromatosis is phlebotomy, which can have unwanted side-effects and which is not an option in cases of secondary hemochromatosis, such as thalassemia, since patients in this case are also anemic (*Brissot et al., 2011*; *Gattermann, 2009*). A potential alternative strategy to counteract excessive iron uptake would be the interference of transport by inhibition of DMT1 (*Crielaard et al., 2017*). Due to the accessibility of the transporter from the apical side, inhibition could proceed from the intestinal lumen by compounds that would not have to cross the membrane.

DMT1 is a member of the conserved SLC11/NRAMP family, which is expressed in all kingdoms of life and which constitutes transporters for transition metals such as $Fe^{2+}$ and $Mn^{2+}$ (*Nevo and Nelson, 2006*; *Shawki et al., 2012*). The transport properties of DMT1 have been characterized in detail by employing electrophysiology and cellular uptake studies, which demonstrated the broad selectivity for transition metals and the discrimination of alkaline earth metal ions such as calcium. The same studies also revealed the symport of $H^+$, which serves as energy source for concentrating the substrate in the cell (*Bozzi et al., 2016a*; *Mackenzie et al., 2006*; *Pujol-Giménez et al., 2017*; *Tandy et al., 2000*).

Insight into the structural basis of transport was provided from different prokaryotic homologues. Crystal structures and functional characterization of these prokaryotic transporters uncovered the general architecture of the protein family and defined the conformational transitions during transport (*Bozzi et al., 2016b*; *Bozzi et al., 2019*; *Ehrnstorfer et al., 2014*; *Ehrnstorfer et al., 2017*). Additionally, these studies defined the chemistry of metal ion coordination and provided insight into the mechanism of proton coupling. SLC11 transporters are monomers and their fold resembles the general organization of a branch of the APC (amino acid-polyamine-cation) superfamily containing transporters for amino acids and glucose (*Forrest and Rudnick, 2009*; *Yamashita et al., 2005*). These proteins consist of two topologically related units of five transmembrane (TM) helices each, which are arranged within the membrane with opposite orientation. The substrate binding site is located in the center of the protein formed by residues in unwound regions of the pseudo-symmetry related α-helices 1 and 6. In metal ion transporters of the SLC11 family the binding-site contains side-chains of conserved aspartate and asparagine residues located on α1 which are directly involved in ion interactions and a methionine residue on α6, which acts as soft ligand that is capable of coordinating transition metal ions but not $Ca^{2+}$ (*Bozzi et al., 2016a*; *Ehrnstorfer et al., 2014*). Large aqueous cavities that alternately provide access to the substrate binding site from opposite sides of the membrane are found in the inward-facing conformation of the transporter from *Staphylococcus capitis* (ScaDMT) (*Ehrnstorfer et al., 2014*), the outward-facing conformation of the transporter from *Eremococcus coleocola* (EcoDMT) (*Ehrnstorfer et al., 2017*) and in multiple structures of the transporter from *Deinococcus radiodurans* (DraDMT), which occupy different states on the transport cycle (*Bozzi et al., 2016b*; *Bozzi et al., 2019*).

Although different structural and functional studies have revealed the mechanism of transition metal ion transport, the pharmacology of SLC11 transporters is still poorly characterized, which has thus far prevented the therapeutic exploration of DMT1 inhibition. The most potent inhibitors of DMT1 identified by screening of large synthetic libraries (*Buckett and Wessling-Resnick, 2009*; *Cadieux et al., 2012*; *Montalbetti et al., 2015*; *Zhang et al., 2012*) are aromatic bis-isothiourea substituted compounds, which display $IC_{50}$ values in the low micromolar range and presumably work by a competitive mechanism (*Montalbetti et al., 2015*; *Zhang et al., 2012*). Whereas studies in a rat model of iron hyperabsorption showed reduced iron uptake in the presence of these inhibitors, underlining the general validity of the approach (*Zhang et al., 2012*), their binding mode to the protein has remained elusive. To overcome this bottleneck in our mechanistic understanding of inhibition and aid the improvement of inhibitors by structure-based design, we have here characterized the detailed interactions between aromatic bis-isothiourea based compounds and their derivatives with human DMT1 and its prokaryotic homologue EcoDMT. Our study combines chemical synthesis with data from X-ray crystallography, isothermal titration calorimetry, in vitro transport and cellular uptake studies to demonstrate that the characterized inhibitors interact with pro- and eukaryotic transporters in a similar manner although with species-dependent differences. These compounds bind deep in a funnel-shaped cavity leading to the metal ion coordination site with one of the isothiourea groups directly interacting with residues of this site thus interfering with substrate loading and locking the transporter in its outward-facing conformation.

## Results

### Functional characterization of the interaction of bis-isothiourea substituted aromatic compounds with human DMT1

To characterize the inhibition mechanism of human DMT1 and its prokaryotic homologue EcoDMT by bis-isothiourea-containing aromatic compounds, we have synthesized seven molecules of this

substance class. These include five compounds carrying two isothiourea moieties for which we have varied the aromatic scaffolds (*i.e.* a brominated dibenzofuran and a single phenyl ring with different substituents) to investigate the influence of their respective size and geometry on inhibition (*Figure 1A*, Appendix 1). For simplicity, we termed the tri-methyl and tri-ethyl substituted benzyl bis-isothiourea compounds TMBIT and TEBIT, respectively, and the dibenzofuran-based compound Br-DBFIT. Br-DBFIT, TMBIT and its derivatives were previously described as inhibitors of DMT1 (*Zhang et al., 2012*). To ease the identification of benzyl bis-isothiourea compounds in inhibitor complexes by X-ray crystallography, we have also synthesized the brominated derivatives Br-BIT and oBr-BIT. Additionally, we have synthesized two variants of the inhibitor oBr-BIT where we have replaced one or both isothiourea moieties by bulkier thio-2-imidazoline groups. All molecules are water-soluble and thus poorly membrane-permeable with both basic isothiourea groups being

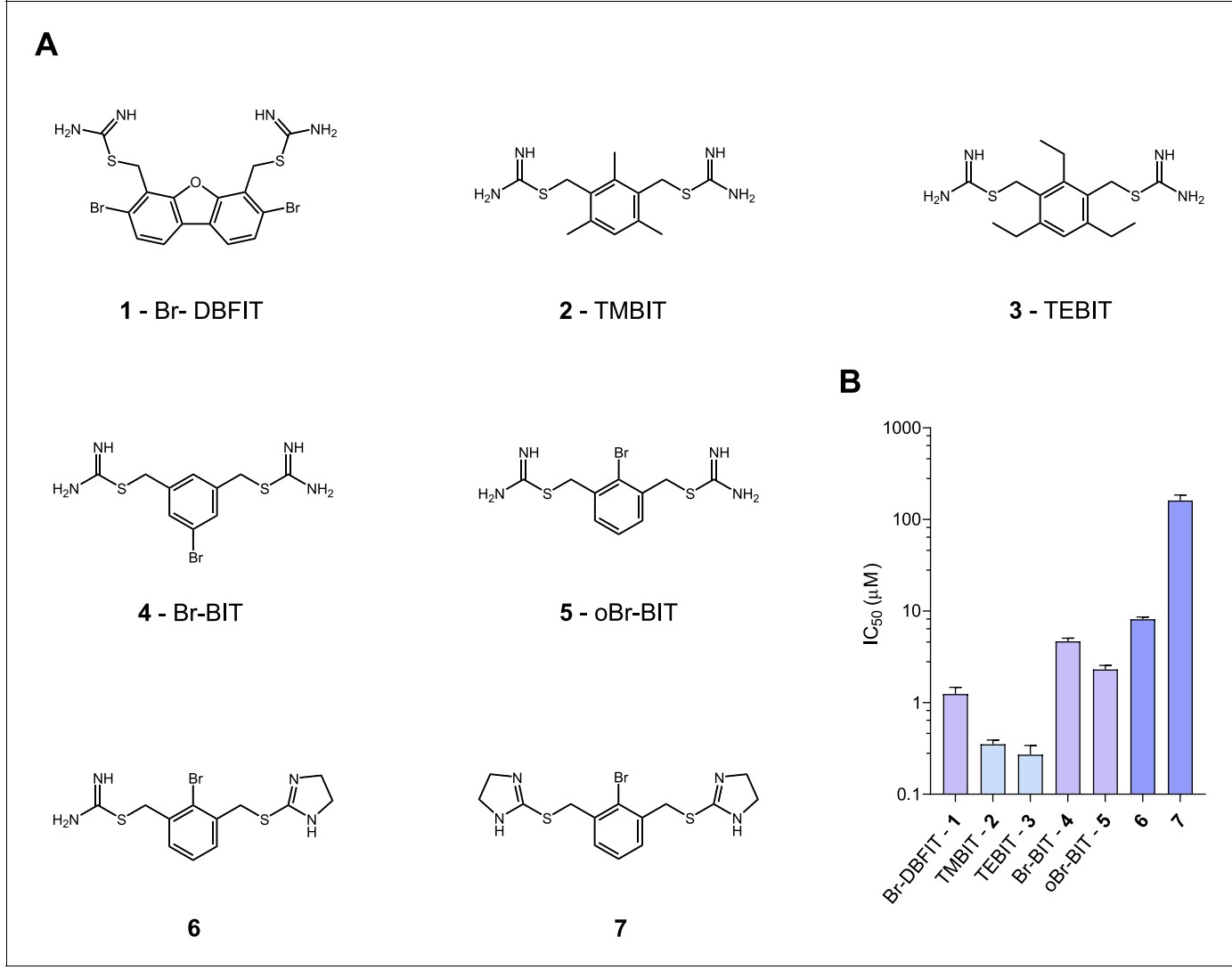

**Figure 1.** Chemical structure of compounds and their inhibition of human DMT1. (**A**) Chemical structures of compounds used in this work. Their synthesis is described in detail in Appendix 1. Abbreviations are indicated. (**B**) $IC_{50}$ values determined by measuring radioactive $^{55}Fe^{2+}$ transport (at 1 μM) into HEK293 cells stably expressing hDMT1. Data from brominated compounds are colored in lilac and data from compounds with modified isothiourea groups in violet. Values show averages of 6–8 biological replicates, errors are s.d.

The online version of this article includes the following figure supplement(s) for figure 1:

**Figure supplement 1.** Characterization of compounds.
**Figure supplement 2.** Inhibition of hDMT1.
**Figure supplement 3.** Transport kinetics of hDMT1 at different inhibitor concentrations.

predominantly charged under physiological conditions (pKa = 8.5–9.5 as measured in a titration of TMBIT and Br-BIT, *Figure 1—figure supplement 1A*). We first tested the activity of all compounds on human DMT1 (hDMT1) by measuring radioactive $^{55}Fe^{2+}$ transport into HEK293 cells stably expressing the protein. When assayed at a free $Fe^{2+}$ concentration of 1 µM, all compounds inhibit metal ion uptake in a dose-dependent manner with $IC_{50}$ values in the micromolar range (*Figure 1B*, *Figure 1—figure supplement 2*). The most potent compounds TEBIT and TMBIT display $IC_{50}$ values of 0.27 µM and 0.35 µM, respectively, latter being in close quantitative agreement with a previous measurement using a calcein-based fluorescence assay ($IC_{50}$ = 0.29 µM) (*Figure 1B*, *Figure 1—figure supplement 2B,C*) (*Zhang et al., 2012*). In comparison, the larger values of Br-BIT (4.66 µM) and oBr-BIT (2.3 µM) indicate an equivalent interaction with somewhat lower affinity and the dibenzofuran compound Br-DBFIT (1.24 µM) is in our hands less potent than previously reported (*Figure 1B*, *Figure 1—figure supplement 2A,D and E*) (*Zhang et al., 2012*). To rule out that the observed activity would be due to chelation of divalent metal ions, we performed isothermal titration calorimetry measurements. Upon titrating $MnCl_2$ to either TMBIT or Br-BIT, we did not detect any pronounced response that would indicate specific binding (*Figure 1—figure supplement 1B*), emphasizing that the inhibition of $^{55}Fe^{2+}$ transport is caused by the specific interactions of either compound with hDMT1. We next investigated the role of the positively charged isothiourea groups for protein interactions by comparing the potency of oBr-BIT with its variants where one or two of the moieties were replaced. In case of the replacement of a single isothiourea group, we were able to measure a four-fold reduced potency of 8.13 µM whereas a much stronger reduction ($IC_{50}$ = 161 µM) was obtained for a compound where both isothioureas were modified (*Figure 1—figure supplement 2F,G*). Together these results underline the importance of the isothiourea moieties for specific protein interactions. To further characterize the mode of inhibition, we studied the effect of different extracellular inhibitor concentrations on the kinetics of iron transport (*Figure 1—figure supplement 3*). In the absence of inhibitors, the transport rate at different $^{55}Fe^{2+}$ concentrations can be fitted to a Michaelis-Menten equation with $K_M$ values of 2.6 µM to 4.4 µM and $v_{max}$ values of 2.7 to 6.1 pmol min$^{-1}$ well$^{-1}$, which is in general agreement with previously reported values (*Gunshin et al., 1997*; *Mackenzie et al., 2006*; *Pujol-Giménez et al., 2017*). At increasing inhibitor concentrations, we observed in all tested cases a pronounced increase of the apparent $K_M$ whereas the apparent $v_{max}$ values decreased only slightly (*Figure 1—figure supplement 3*, *Table 1*). These results suggest that the compounds act by a predominant competitive mechanism. When fitting the data to a mixed enzyme inhibition model the resulting equilibrium constants are in the micromolar range with inhibitors binding with much higher affinity to the substrate-free transporter (*Table 1*). Taken together, our data confirm the activity of aromatic isothiourea-based compounds as competitive inhibitors of hDMT1 with potencies in the low micromolar range. As all compounds are positively charged and thus membrane-impermeable, the binding site of the inhibitor is expected to be accessible from the extracellular side.

## Functional characterization of the interaction with EcoDMT

After the characterization of hDMT1 inhibition, we have studied the properties of different bis-isothiourea compounds on the prokaryotic SLC11 homologue EcoDMT, which catalyzes $H^+$-coupled $Mn^{2+}$ symport and whose structure was determined in an outward-facing conformation by X-ray crystallography (*Ehrnstorfer et al., 2017*). Due to the insufficient solubility of the dibenzofuran-based inhibitor Br-DBFIT and TEBIT for experiments with EcoDMT, we restricted this analysis to the benzyl bis-isothiourea compounds TMBIT, Br-BIT and oBr-BIT. To characterize EcoDMT-mediated transport, we have reconstituted the purified protein into liposomes and used a fluorescence-based in-vitro assay (*Figure 2—figure supplement 1A*). In these proteoliposomes, EcoDMT is incorporated in inside-out and outside-out orientations at about equal ratios (*Figure 2—figure supplement 1B*). Concentration-dependent $Mn^{2+}$ uptake into proteoliposomes was monitored by the time-dependent quenching of the fluorophore calcein trapped inside the vesicles (*Figure 2—figure supplement 1A*) (*Ehrnstorfer et al., 2017*). In the absence of inhibitors, $Mn^{2+}$ transport by EcoDMT saturates at low micromolar concentrations ($K_M$ = 4.3 µM) (*Figure 2—figure supplement 1C*, *Table 1*). The addition of either benzyl bis-isothiourea compound decreases the kinetics of uptake in a dose-dependent manner thus suggesting that all tested compounds, when applied at micromolar concentrations to the outside of proteoliposomes, inhibit the transport activity of EcoDMT by binding to a saturable site of the protein (*Figure 2A*, *Figure 2—figure supplement 1D,E*). Since higher concentrations of

**Table 1.** Transport kinetics and inhibition.

| Protein | Inhibitor | $K_M$ (μM) | $v_{max}$* | $K_i$ (μM) | α |
|---|---|---|---|---|---|
| | | *Michaelis-Menten* | | *Equation 2* | |
| hDMT1 | - | 3.1 ± 0.6 | 4.6 ± 0.3 | 0.57 ± 0.17 | >100 |
| | 1 μM Br-DBFIT | 9.2 ± 1.2 | 4.2 ± 0.2 | | |
| | 2.5 μM Br-DBFIT | 14.7 ± 2.8 | 4.3 ± 0.4 | | |
| | 5 μM Br-DBFIT | 14.4 ± 3.0 | 3.4 ± 0.4 | | |
| hDMT1 | - | 2.6 ± 0.5 | 2.6 ± 0.1 | 0.35 ± 0.07 | 17.4 |
| | 0.25 μM TMBIT | 4.0 ± 0.4 | 2.3 ± 0.1 | | |
| | 0.5 μM TMBIT | 6.1 ± 0.8 | 2.4 ± 0.1 | | |
| | 1.25 μM TMBIT | 8.0 ± 1.0 | 2.1 ± 0.1 | | |
| hDMT1 | - | 2.9 ± 0.4 | 3.1 ± 0.1 | 0.08 ± 0.01 | >100 |
| | 0.25 μM TEBIT | 13.1 ± 3.3 | 3.0 ± 0.4 | | |
| | 0.5 μM TEBIT | 18.3 ± 5.3 | 2.9 ± 0.5 | | |
| | 1.25 μM TEBIT | 26.8 ± 8.6 | 2.5 ± 0.5 | | |
| hDMT1 | - | 2.6 ± 0.5 | 2.6 ± 0.1 | 3.6 ± 0.7 | 13.5 |
| | 5 μM Br-BIT | 6.7 ± 0.7 | 2.4 ± 0.1 | | |
| | 10 μM Br-BIT | 7.4 ± 1.2 | 2.2 ± 0.1 | | |
| | 25 μM Br-BIT | 12.4 ± 3.7 | 1.6 ± 0.2 | | |
| hDMT1 S476V | - | 2.3 ± 0.5 | - | - | - |
| hDMT1 N520L | - | 2.4 ± 0.5 | - | - | - |
| hDMT1 F523A | - | 2.7 ± 0.6 | - | - | - |
| EcoDMT | - | 4.3 ± 0.5 | 21.9 ± 0.5 | 14.2 ± 2.6 | >100 |
| | 10 μM Br-BIT | 13.7 ± 1.2 | 21.7 ± 0.4 | | |
| | 50 μM Br-BIT | 21.7 ± 3.6 | 21.4 ± 1.0 | | |
| | 100 μM Br-BIT | 31.5 ± 6.9 | 20.4 ± 1.3 | | |
| EcoDMT N456A | - | 12.5 ± 1.7 | 19.1 ± 0.6 | 29.3 ± 7.1 | 8.5 |
| | 10 μM Br-BIT | 17.8 ± 4.0 | 18.3 ± 1.1 | | |
| | 50 μM Br-BIT | 28.8 ± 5.4 | 14.9 ± 0.8 | | |
| | 100 μM Br-BIT | 33.9 ± 6.9 | 13.8 ± 0.8 | | |
| EcoDMT N456L | - | 7.5 ± 0.7 | 17.3 ± 0.4 | 28.8 ± 5.3 | 29.2 |
| | 10 μM Br-BIT | 13.1 ± 1.7 | 16.2 ± 0.5 | | |
| | 50 μM Br-BIT | 19.0 ± 2.4 | 13.8 ± 0.5 | | |
| | 100 μM Br-BIT | 27.0 ± 3.9 | 15.1 ± 0.6 | | |
| EcoDMT N456A S459A Q463A | | 7.2 ± 0.9 | 21.9 ± 0.5 | 23.8 ± 3.2 | >100 |
| | 10 μM Br-BIT | 15.7 ± 1.7 | 21.7 ± 0.4 | | |
| | 50 μM Br-BIT | 32.3 ± 4.5 | 21.4 ± 1.0 | | |
| | 100 μM Br-BIT | 47.9 ± 5.6 | 20.4 ± 1.3 | | |

*$v_{max}$ values for hDMT1 measurements are given in pmol min$^{-1}$ well$^{-1}$ and vmax values for EcoDMT measurements in ΔF Δt$^{-1}$.

TMBIT and oBr-BIT (i.e. >50 μM) did interfere with the assay, we restricted our quantitative characterization to Br-BIT, where we do not observe any interference at concentrations up to 200 μM. At high micromolar concentrations of Br-BIT, the decrease of transport activity approaches a maximum and even at 200 μM Br-BIT we could not detect complete inhibition. The saturation of the inhibition at high concentration results from the full occupancy of accessible binding sites, whereas the residual transport likely originates from transporters with inside-out orientation which do not expose the presumed inhibitor binding site to the external solution. The basal activity at high inhibitor

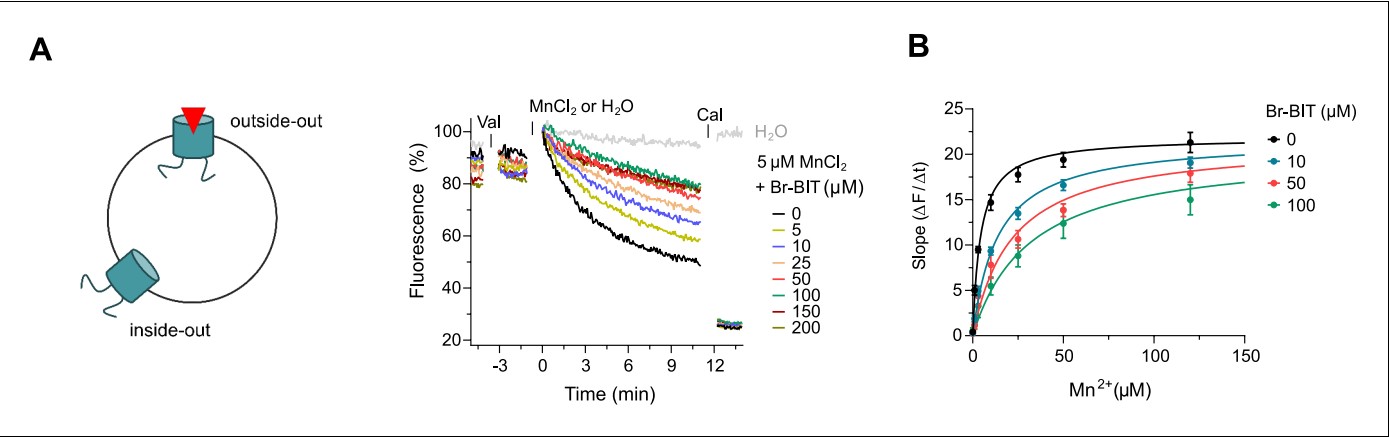

**Figure 2.** Inhibition of EcoDMT by Br-BIT. (**A**) EcoDMT mediated $Mn^{2+}$ transport into proteoliposomes in presence of Br-BIT assayed by the quenching of the fluorophore calcein trapped inside the vesicle. Br-BIT was added at different concentration (5–200 µM) to the outside of the liposomes. The transport is started by the addition of the potassium ionophore Valinomycin (Val) and 5 µM $MnCl_2$. To equilibrate the $Mn^{2+}$ ions, the ionophore Calcimycin (Cal) was added at the end of the run. Schematic figure (left) depicts the sidedness of inhibition which is responsible for the remaining activity at large inhibitor concentrations. (**B**) Transport kinetics of EcoDMT at the indicated Br-BIT concentrations. The solid lines are fits to the Michaelis-Menten equation assuming similar kinetic properties of transport for both orientations of the transporter. The observed kinetic parameters thus describe apparent values obtained from an average of transport properties in inside-out and outside-out orientations. Values show averages of six technical replicates obtained from three independent proteoliposome preparations, errors are s.e.m.

The online version of this article includes the following figure supplement(s) for figure 2:

**Figure supplement 1.** Characterization of EcoDMT inhibition.

concentration thus further demonstrates the sidedness of the inhibition and the membrane-impermeability of the compound. As for the inhibition of hDMT1, the $K_M$ values of transport increased at higher inhibitor concentrations, whereas $v_{max}$ did not show pronounced changes (*Figure 2B*, *Table 1*). The $K_i$ value of 14.2 µM, representing the equilibrium dissociation constant to the substrate-free EcoDMT, is in the same range as the $K_i$ value of 3.6 µM obtained for hDMT1, reflecting the strong structural relationship between both proteins. Together, our results suggest that Br-BIT inhibits EcoDMT and hDMT1 by a common competitive mechanism.

## Structural characterization of the inhibition of EcoDMT by Br-BIT

To investigate the structural basis for the inhibition of divalent metal ion transporters of the SLC11 family by benzyl bis-isothiourea-based compounds, we have characterized the interaction between the brominated analogs and EcoDMT by X-ray crystallography. In our experiments we exploited the anomalous scattering properties of the inhibitors to facilitate their localization in the complex. For that purpose, we have soaked crystals of EcoDMT with Br-BIT and oBr-BIT and collected multiple datasets at a wavelength corresponding to the anomalous absorption edge of bromine (*Table 2*, *Table 3*). Whereas we were unable to detect bromine in the anomalous maps of oBr-BIT containing crystals, the majority of datasets collected from crystals soaked with Br-BIT displayed a single strong peak in the anomalous difference density at equivalent positions, which aided the localization of the bound inhibitor (*Figure 3A,B*, *Figure 3—figure supplement 1A*, *Table 3*). A detailed view of the complex defined in the 2Fo-Fc density at 3.8 Å is displayed in *Figure 3A*. In this structure, EcoDMT adopts the same substrate-free outward-facing conformation that has previously been observed in datasets of the protein in absence of the inhibitor (*Figure 3C,D*) (*Ehrnstorfer et al., 2017*). In this conformation, a funnel-shaped aqueous pocket of the protein leads from the extracellular solution to the substrate binding site. The inhibitor is bound at the base of this pocket as defined by the anomalous difference density that constrains the position of the covalently bound Br-atom and by residual density in the 2Fo-Fc omit map that was calculated with phases from a model not containing the inhibitor (*Figure 3B*). The fact that the Br-Atom of Br-BIT is located in the narrow apex of the pocket, whereas it would be placed in the wider part of the cavity in oBr-BIT might explain why we were unable to detect the binding position in the anomalous difference density of the latter compound. The omit map of the EcoDMT Br-BIT complex displays density for the aromatic ring and for

**Table 2.** X-ray data collection and refinement statistics.

| | EcoDMT-Br-BIT complex |
|---|---|
| **Data collection** | |
| Space group | C2 |
| Cell dimensions | |
| $a, b, c$ (Å) | 150.0, 81.7, 95.6 |
| $\alpha, \beta, \gamma$ (°) | 90, 107.4, 90 |
| Wavelength (Å) | 0.92 |
| Resolution (Å) | 50–3.8 (3.9–3.8)[*] |
| $R_{merge}$ (%) | 9.8 (154.6) |
| $CC_{1/2}$ (%) | 100.0 (81.7) |
| $I/\sigma I$ | 15.6 (2.1) |
| Completeness (%) | 99.3 (99.3) |
| Redundancy | 14.3 (13.9) |
| **Refinement** | |
| Resolution (Å) | 12–3.8 |
| No. Reflections | 10576 |
| $R_{work}/R_{free}$ (%) | 21.6/25.8 |
| No. atoms | |
| Protein | 3780 |
| Ligand/ion | 17 |
| Water | - |
| *B* factors | |
| Protein | 171.3 |
| Ligand/ion | 237.1 |
| r.m.s. deviations | |
| Bond lengths (Å) | 0.005 |
| Bond angles (°) | 0.65 |

[*]Values in parentheses are for highest-resolution shell.

the isothiourea group located close to the metal ion binding site (termed proximal isothiourea group), whereas the other group (the distal isothiourea group) is not defined in the electron density reflecting its increased conformational flexibility (*Figure 3B*). In general, the shape of the binding pocket is complementary to the structure of the inhibitor but it is sufficiently wide in the long direction of the molecule to accommodate substitutions at the aromatic ring as found in the molecules TMBIT, TEBIT and in the larger dibenzofuran ring of Br-DBFIT (*Figure 3E*). The aromatic group is stacked between α-helices 6 and 10 contacted by the side chains of residues Ala 231, Leu 410, Ala 409 and Leu 414. The close-by Asn 456 located on α11 might interact with the covalently attached Br atom of Br-BIT (*Figure 3F*, *Figure 3—figure supplement 1B*). The proximal isothiourea group is located in a narrow pocket in interaction distance to the conserved Asp 51 and Asn 54 in the unwound part of α−1 and to Gln 407 on α−10, which were shown to contribute to the coordination of transported metal ions (*Figure 3E,F*, *Figure 3—figure supplement 1B*) (*Bozzi et al., 2019*; *Ehrnstorfer et al., 2014*; *Ehrnstorfer et al., 2017*; *Pujol-Giménez et al., 2017*). The distal isothiourea group is located in the wider entrance of the cavity and might thus adopt different conformations, which is consistent with its undefined position in the electron density (*Figure 3B*). In one conformation, this group approaches residues Ser 459 and Gln 463, both located on α11. Besides the direct ionic interactions of the proximal isothiourea group with Asp 51, the positive charge of both groups would be additionally stabilized by the negative electrostatics of the pocket that is conferred by an excess of acidic residues (*Figure 3—figure supplement 1C*). The observed binding

**Table 3.** X-Ray data collection statistics of additional datasets.

| EcoDMT-Br-BIT complexes | 1 | 2 | 3 | 4 | 5 |
|---|---|---|---|---|---|
| **Data collection** | | | | | |
| Space group | C2 | C2 | C2 | C2 | C2 |
| Cell dimensions (Å), (°) | | | | | |
| a b c | 147.9 81.2 95.3 | 149.7 81.7 95.5 | 148.7 81.1 94.8 | 150.1 81.6 95.5 | 148.9 81.3 95.4 |
| α β γ | 90 107.3 90 | 90 107.3 90 | 90 107.1 90 | 90 107.3 90 | 90 107.3 90 |
| Wavelength (Å) | 0.92 | 0.92 | 0.92 | 0.92 | 0.92 |
| Resolution (Å) | 50–3.8 (3.9–3.8)* | 50–4.0 (4.1–4.0)* | 50–4.3 (4.4–4.2)* | 50–4.1 (4.2–4.1)* | 50–4.2 (4.3–4.2)* |
| $R_{merge}$ (%) | 6.7 (182.1) | 7.8 (142.0) | 7.2 (152.6) | 8.3 (112.5) | 11.8 (198.9) |
| $CC_{½}$ (%) | 100.0 (80.4) | 100.0 (81.0) | 100.0 (71.6) | 100.0 (53.9) | 100.0 (77.1) |
| $I/\sigma I$ | 15.6 (1.5) | 19.3 (1.6) | 14.8 (1.4) | 13.7 (1.6) | 14.5 (1.5) |
| Completeness (%) | 99.2 (99.7) | 99.0 (99.5) | 97.4 (74.0) | 94.8 (40.5) | 99.8 (100.0) |
| Redundancy | 14.6 (14.6) | 21.0 (11.7) | 14.2 (11.8) | 8.7 (5.1) | 14.0 (14.4) |

*Values in parentheses are for highest-resolution shell.

position and the assumed interaction of the inhibitor with the metal ion binding site is also compatible with the observed competitive nature of the inhibition.

The high sequence similarity between bacterial and human orthologs (i.e. 52% similar and 29% identical residues between EcoDMT and hDMT1) facilitates the construction of a homology model of human DMT1 (*Figure 3—figure supplement 2A,B*), which permits a glimpse of potential interactions of the inhibitor with the human transporter. As this model does not contain any insertions or deletions in the binding region, we expect a similar-shaped outward-facing cavity binding the inhibitor in hDMT1 as observed for EcoDMT (*Figure 3G*, *Figure 3—figure supplement 2B*). The conservation is particularly high for α-helices 1 and 6 constituting the metal ion coordination site, but differences are observed for pocket-lining residues located on α-helices 10 and 11: While the corresponding residues Leu 414 in EcoDMT and Leu 479 in hDMT1 (both located on α10) seal the bottom of the binding cavity in both proteins, the hydrophobic character of Leu 410 and Ala 409 in EcoDMT, which contact one face of the aromatic ring is altered by the polar sidechains of Gln 475 and Ser 476 in hDMT1 (*Figure 3F,H*, *Figure 3—figure supplement 2A*). Close to the ion coordination site, Gln 407 of EcoDMT, which potentially interacts with the proximal isothiourea group, is substituted by an Asn 472 in hDMT1 (*Figure 3F,H*). Among the α-helices constituting the inhibitor binding site, α−11 is least conserved between hDMT1 and EcoDMT (*Figure 3—figure supplement 2A*). Whereas Asn 456 of EcoDMT, which contacts the Br atom of Br-BIT is conserved in hDMT1 (Asn 520), Ser 459 and Gln 463 of the bacterial transporter are replaced by bulky aromatic residues (Phe 523 and Tyr 527), which could decrease the volume of the binding pocket in hDMT1 and thus potentially constrain the conformation of the distal isothiourea group (*Figure 3F,G,H*). Nevertheless, since both residues are located in the wider part of the binding pocket, it is justified to assume a similar general binding mode of the inhibitor in bacterial and human orthologues. As for EcoDMT, we expect that the strongly negative electrostatic potential within the binding pocket of hDMT1 favors the binding of the positively charged inhibitor (*Figure 3—figure supplement 2C*). Taken together, our structural data thus provide a detailed view of the molecular basis of the interaction of benzyl bis-isothiourea-based inhibitors with divalent metal ion transporters of the SLC11/NRAMP family.

## Functional characterization of inhibitor binding-site mutants of EcoDMT

To further characterize the binding of Br-BIT to EcoDMT, we have studied the effect of mutations of putative contact residues identified in the structure on inhibition (*Figure 4A*). Although the described results emphasize the importance of interactions of the isothiourea group with the metal-

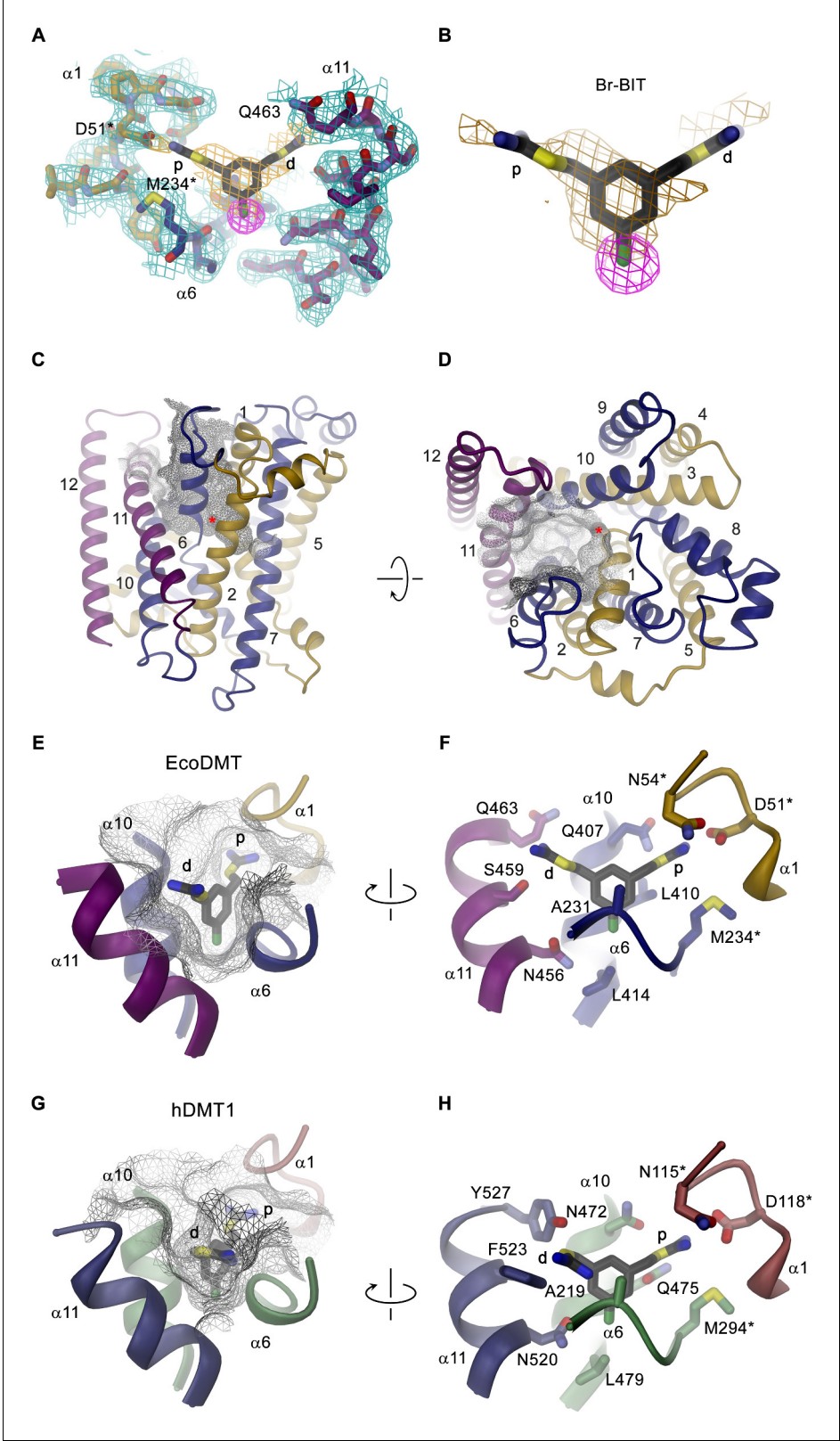

**Figure 3.** Structure of inhibitor complexes. (A-B) Close-up views of the crystal structure of EcoDMT in complex with Br-BIT determined at 3.8 Å resolution viewed from within the membrane. The refined $2F_o$-$F_c$ electron density is shown as blue mesh. The position of Br-BIT is defined by the anomalous difference density of the Br-atom (shown as magenta mesh, contoured at 7σ) and by residual density in the $2F_o$-$F_c$ omit map (dark yellow mesh). *Figure 3 continued on next page*

*Figure 3 continued*

C-D, Ribbon representations of the EcoDMT structure viewed from within the membrane (**C**) and from the extracellular side (**D**). The $Mn^{2+}$ binding site is indicated with a red asterisk. The molecular surfaces are represented as gray meshes. The C-terminal sub-domain ($\alpha$-helices 6–12) is shown in dark blue, $\alpha$-helices 11 and 12 in magenta. (**E**) Position of Br-BIT in the binding pocket (gray mesh) of EcoDMT. (**F**) Detailed view of the residues in contact distance to Br-BIT. (**G**) Position of Br-BIT in the binding pocket (gray mesh) of a homology model of human DMT1. (**H**) Potential interactions of Br-BIT with the homology model of human DMT1. A-H, The proximal (**p**) isothiourea group is close to the metal ion coordinating residues (marked with a black asterisk) and the distal (**d**) isothiourea group is in proximity to $\alpha$-helix 11.

The online version of this article includes the following figure supplement(s) for figure 3:

**Figure supplement 1.** Inhibitor binding to EcoDMT.
**Figure supplement 2.** Homology model of hDMT1.

coordination site, these cannot be probed with the applied transport assays as mutations of coordinating resides interfere with ion uptake. We have thus employed isothermal titration calorimetry (ITC) to directly measure the effect of a metal-binding site mutant in EcoDMT on inhibitor binding.

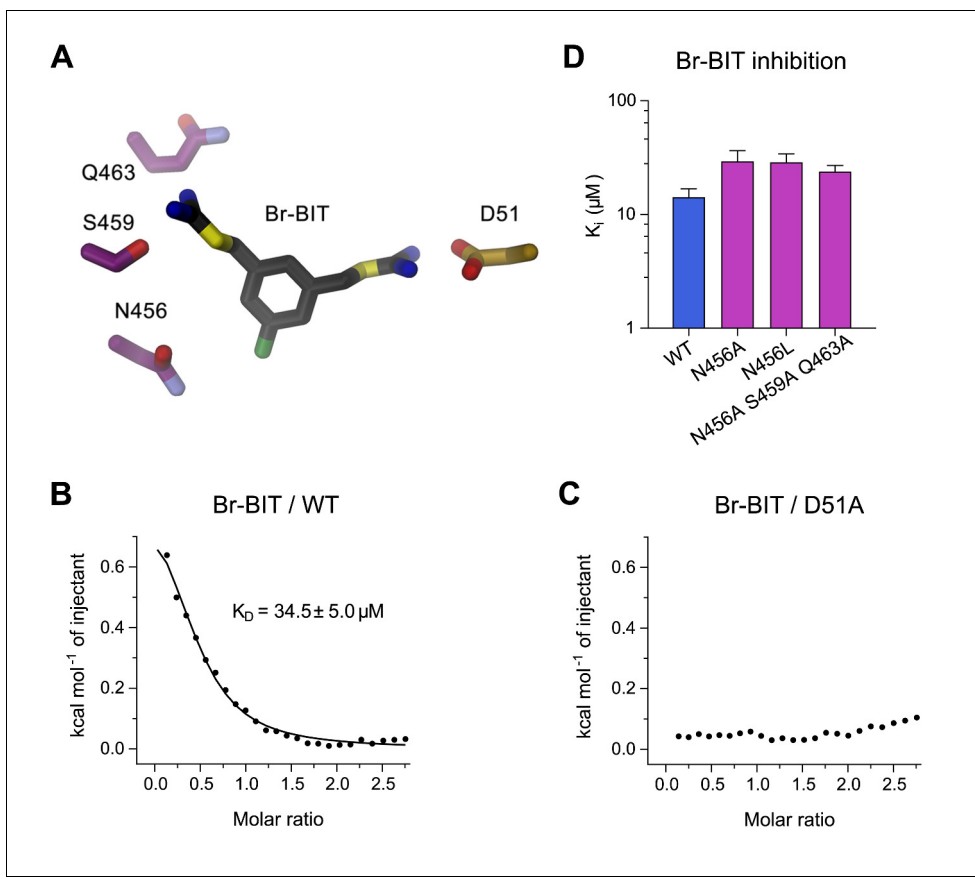

**Figure 4.** Characterization of inhibitor interactions in EcoDMT mutants. (**A**) Close-up view of mutated residues surrounding Br-BIT in EcoDMT. B-C, Binding isotherms obtained from isothermal titrations of Br-BIT to EcoDMT WT (**B**) and the metal ion binding site mutant D51A (**C**). For WT, the data was fitted to a model assuming a single set of binding sites with the binding isotherm shown as solid line. Error represent fitting errors. (**D**) Equilibrium dissociation constants (Ki) of Br-BIT binding to WT and mutants of EcoDMT determined using the proteoliposome transport assay and by fitting the data to the mixed enzyme inhibition model (*Equation 2*).

The online version of this article includes the following figure supplement(s) for figure 4:

**Figure supplement 1.** Isothermal titration calorimetry.
**Figure supplement 2.** Transport inhibition of EcoDMT mutants.

In ITC experiments, we find two signals in the thermograms in response to the titration of the inhibitor to the WT protein. A weak endothermic contribution, which saturates at low µM concentrations ($K_D = 34.5 \pm 5.0$ µM) can be attributed to the loading of the inhibitor binding site and an exothermic signal saturating with an affinity in the mM range to a potential non-specific interaction with the protein (*Figure 4B*, *Figure 4—figure supplement 1A,B*). To characterize the observed interaction between the positively charged isothiourea group and the negatively charged Asp 51 of the metal-binding site, we have expressed and purified the mutant D51A and measured inhibitor binding. Whereas the low-affinity signal in the thermograms appears unaltered, the high affinity component is absent, as expected if the mutant has removed an important interaction which interferes with inhibitor binding (*Figure 4C*, *Figure 4—figure supplement 1A,B*). Thus, despite the weak signal originating from the low enthalpic contribution to binding, our titration calorimetry experiments indicate a direct interaction of the isothiourea group with the metal binding site as expected for a competitive inhibitor.

To probe the role of other residues of EcoDMT in the vicinity of the bound inhibitor, we have characterized the effect of alterations of three hydrophilic residues on α-helix 11 on the inhibition of $Mn^{2+}$ transport. Based on our structures, we suspected Gln 463 and Ser 459 to interact with the distal isothiourea group and Asn 456 with the bromine atom on the aromatic ring of Br-BIT (*Figure 4A*). The three constructs, the single mutants N456A and N456L and the triple mutant N456A/S459A/Q463A transport $Mn^{2+}$ with similar kinetics as WT (*Figure 4—figure supplement 2A–C*, *Table 1*). As for WT, $Mn^{2+}$ transport in all three mutants is inhibited upon addition of Br-BIT, although with slightly decreased potency ($K_i$: WT, 14.2 µM; N456A, 29.3 µM; N456L, 28.8 µM; N456A/S459A/Q463A, 23.8 µM) (*Figure 4D*, *Figure 4—figure supplement 2D–F*, *Table 1*). In light of the small difference in $K_i$ compared to WT, our data excludes a large energetic contribution of residues on α11 to inhibitor binding, consistent with the assumed mobility of the distal isothiourea group that is manifested in the lack of electron density of the group in the structures of EcoDMT Br-BIT complexes.

## Functional characterization of inhibitor binding-site mutants of hDMT1

To characterize the role of residues in the predicted inhibitor binding pocket of human DMT1, we have generated several point mutants and investigated the effect of these mutations on the interaction with different inhibitors. Due to the strong negative impact of alterations of the metal ion coordination site on transport, mutagenesis was restricted to residues lining the remainder of the binding pocket. The investigated positions encompassed residues on α−6 (Ala 291), α−10 (Gln 475, Ser 476 and Leu 479), and α−11 (Asn 520, Phe 523 and Tyr 527) (*Figures 3H* and *5A*). In our experiments we wanted to target interactions of protein residues with the aromatic ring in the narrow part of the binding pocket by either shortening the side-chains in the mutants A291G and Q475A, or by increasing their size in the mutants A291V, Q475F, S476V and L479F. In the orthogonal direction, the binding pocket is wider and would on one side be delimited by resides located on α−11 (*Figures 3G* and *5A*). Based on our model, we suspected the aromatic side chains of Phe 523 and Tyr 527 to be located in proximity to the distal isothiourea groups of TMBIT, TEBIT and Br-BIT or to the second phenyl-ring in the case of the dibenzofuran-based compound Br-DBFIT and Asn 520 in interaction distance with the aromatic ring harboring the proximal isothiourea group in all compounds (*Figures 3H* and *5D*). To probe these potential interactions, we have truncated the aromatic side chains in the mutants F523A and Y527A and generated a nearly isosteric hydrophobic substitution in the mutant N520L and subsequently studied the $^{55}Fe^{2+}$ uptake properties of HEK293 cells transiently transfected with DNA coding for the respective constructs. Transport is similar to WT in case of the mutants S476V, F523A and Y527A, reduced in the mutants Q475A and N520L and undetectable in the mutants A291G, A291V, Q475F and L479F (*Figure 5B*, *Figure 5—figure supplement 1A*). Mutations that render hDMT1 inactive, most likely interfere with structural rearrangements during ion transport, as judged by the tight packing of the respective region in the inward-facing structures of SLC11 transporters (*Bozzi et al., 2016b*; *Bozzi et al., 2019*; *Ehrnstorfer et al., 2014*). Inhibition experiments on hDMT1 were carried out with Br-BIT used for crystallization, the more potent inhibitors TMBIT and TEBIT and the dibenzofuran-based compound Br-DBFIT to explore the influence of the aromatic scaffold and the geometric relationship between the two isothiourea groups on interactions. Similar to WT, the addition of either compound at equivalent concentrations decreases uptake in the mutants Y527A and Q475A both located towards the extracellular entrance

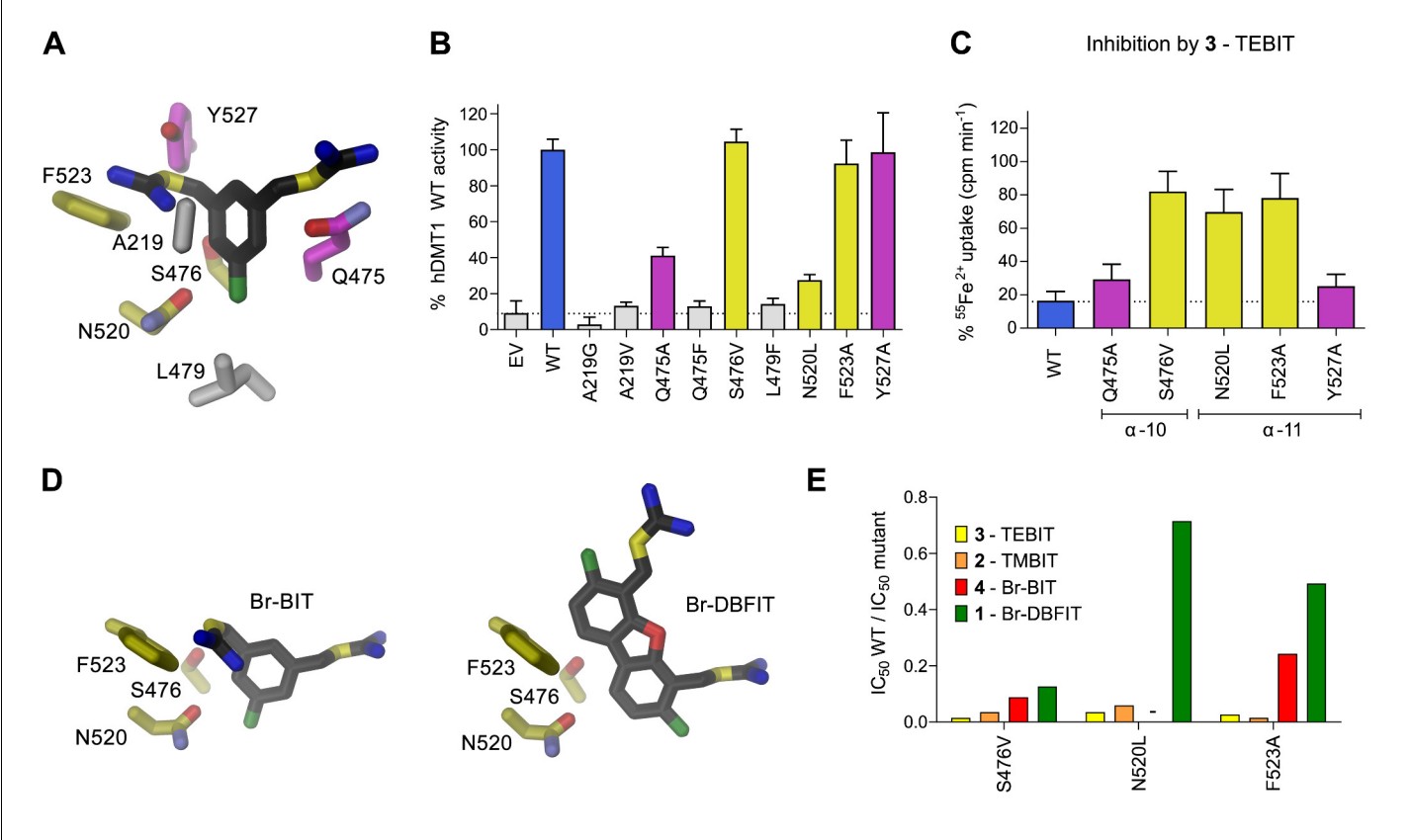

**Figure 5.** Characterization of inhibitor interactions in hDMT1 mutants. A, D Close-up view of mutated residues surrounding Br-BIT (A, D) and Br-DBFIT (D) in a homology model of hDMT1. (B) Transport activity of HEK293 cells transiently transfected with a vector not containing an insert (EV) or with WT and human DMT1 mutants determined using a cellular iron uptake assay. Data show mean of 15–39 replicates, errors are s.d. (C) Inhibition of cellular iron uptake by 10 µM TEBIT of HEK293 cells transiently transfected with WT and mutants of human DMT1 at 1 µM $Fe^{2+}$. Data show mean of 6–9 replicates, errors are s.d. (E) Characterization of inhibitory properties of Br-BIT, TMBIT, TEBIT and Br-DBFIT in HEK293 cells transiently transfected with the respective hDMT1 mutants. The ratio of $IC_{50}$ WT / $IC_{50}$ mutant is displayed. Low values, reflect a strong effect of the mutation on inhibition. As Br-BIT did not inhibit transport of the mutant N520L at the tested concentrations, $IC_{50}$ ratios are not displayed (-).

The online version of this article includes the following figure supplement(s) for figure 5:

**Figure supplement 1.** Transport inhibition of hDMT1 mutants.

to the binding pocket (*Figures 3H* and *5A,C* and *Figure 5—figure supplement 1B–D*) thus suggesting that interactions with these residues do not strongly contribute to inhibitor binding. Conversely, the compounds had much smaller effects on the transport activity of cells expressing the mutants S476V, N520L and F523A located deeper in the binding pocket (*Figures 3H* and *5C,D*, *Figure 5—figure supplement 1B–C*) thus suggesting that in these cases, the mutations affected inhibitor interactions. To further characterize the inhibitory properties of the investigated compounds, we have measured uptake at different inhibitor concentrations and found a strong reduction in potency in most cases (*Figure 5E*, *Table 4*). Whereas the effect is uniform in the mutant S476V for all investigated compounds, the mutants N520L, and F523A showed a decreased potency of inhibition for the related molecules Br-BIT, TMBIT and TEBIT but only a slight reduction for Br-DBFIT (*Figure 5E*, *Figure 5—figure supplement 1E*, *Table 4*) indicating that residues on α−11 might form distinct interactions with different inhibitor classes. This is consistent with the wide dimensions of the pocket in that direction that allows for a geometry-dependent placement of the aromatic ring and the attached isothiourea moiety on the distal side (*Figure 5D*). Taken together our results suggest an involvement of residues on α−10 and α−11 on inhibitor binding to hDMT1 although with variable specificity, consistent with the proposed general binding mode of the inhibitors, which constrain the binding of the first aromatic ring to position the proximal isothiourea group in interaction distance with the

**Table 4.** IC$_{50}$ values (µM) of WT and mutant proteins.

| | TEBIT | TMBIT | Br-BIT | Br-DBFIT |
|---|---|---|---|---|
| WT | 0.48 ± 0.09 | 1.28 ± 0.54 | 4.2 ± 1.5 | 1.43 ± 0.34 |
| S476V | 32.1 ± 6.3 | 36.1 ± 8.0 | 47.6 ± 6.8 | 11.3 ± 0.3 |
| N520L | 13.7 ± 3.9 | 21.7 ± 6.9 | - | 2.0 ± 0.4 |
| F523A | 17.8 ± 7.0 | 82.9 ± 5.3 | 17.3 ± 0.5 | 2.9 ± 0.9 |

Measurements were performed using 1 µM Fe$^{2+}$ with transiently transfected HEK293 cells expressing the indicated hDMT1 constructs. The values shown for WT deviate from the values shown in **Figure 1—figure supplement 2**, due to small differences in the experimental setup (*i.e.* the use of a stable cell line *vs.* transiently transfected cells).

metal ion coordination site. Since equivalent mutations of α11 in EcoDMT had little impact on inhibition of Br-BIT, our results also point towards species-dependent energetic differences in inhibitor interactions on the distal side of the inhibitor binding pocket, which are reflected in the poor conservation of residues in α11 and the wide geometry of the pocket in the prokaryotic transporter. Despite the described species-dependent differences, our data is generally consistent with the notion that the characterized compounds inhibit both pro- and eukaryotic transporters by binding to equivalent regions.

## Discussion

By combining chemical synthesis with X-ray crystallography and in vitro binding and transport assays on human DMT1 and its prokaryotic homologue EcoDMT, our study has revealed detailed insight into the inhibition of SLC11 transporters by aromatic bis-isothiourea-based compounds. These compounds inhibit pro- and eukaryotic family members by a predominant competitive mechanism by binding to an outward-facing aqueous cavity leading to the transition metal ion coordination site (*Figures 1*, *2*, *3* and *6*) which prevents substrate loading and the transition to an inward-open conformation of the transporter. We have shown that these compounds do not interact with the reactive transported substrate, which has hampered the identification of specific inhibitors in high-throughput screens (*Figure 1—figure supplement 1B*). We have also shown that these compounds are positively charged and thus poorly membrane permeable and most likely attracted and stabilized by the strong negative electrostatic potential in the outward-facing aqueous cavity (*Figure 1—figure supplement 1A*, *Figure 3—figure supplements 1C* and *2C*). Our structural studies have identified the binding mode of the inhibitors at the base of the funnel-shaped cavity, with the aromatic group snugly fitting into the pocket, thereby positioning the isothiourea group into ideal interaction distance with the aspartate of the transition metal binding site (*Figures 3A,B* and *6B*). Although the characterization of the interaction to the metal ion binding site is experimentally challenging, since mutations at this site interfere with transport (*Bozzi et al., 2019*; *Ehrnstorfer et al., 2014*; *Ehrnstorfer et al., 2017*; *Pujol-Giménez et al., 2017*), it is supported by several observations: First, the interaction of the isothiourea group with the metal ion binding site is displayed in the electron density of the complex (*Figure 3A,B*). Second, the low micromolecular binding affinity of the inhibitor to the prokaryotic transporter EcoDMT observed in titration calorimetry experiments vanishes in a mutant truncating the binding site aspartate (*Figure 4B,C*, *Figure 4—figure supplement 1A,B*)). Third, the interaction underlies the observed competitive mechanism that is shared by all investigated isothiourea-based compounds containing different aromatic substituents (*Figure 1*, *Figure 1—figure supplement 3*), and fourth it underlines the strong requirement for the isothiourea group for potent inhibition. Latter is illustrated by the inhibition of human DMT1 by compounds where either one or both isothiourea groups are modified, leading to moderately reduced potency in the first, and a strongly reduced binding affinity in the second compound (*Figure 1B*, *Figure 1—figure supplement 2E,F,G*). In our proposed inhibition mechanism, the role of the aromatic group in each compound is to position the inhibitor at the base of the predominantly hydrophobic pocket in proximity to the binding site (*Figure 6B*). This is supported by the fact that a mutation in hDMT1 that likely narrows the pocket in this direction (S476V) leads to a reduced potency of inhibition (*Figure 5C,E*, *Figure 5—figure supplement 1B–E*, *Table 4*). In the orthogonal direction, the funnel-

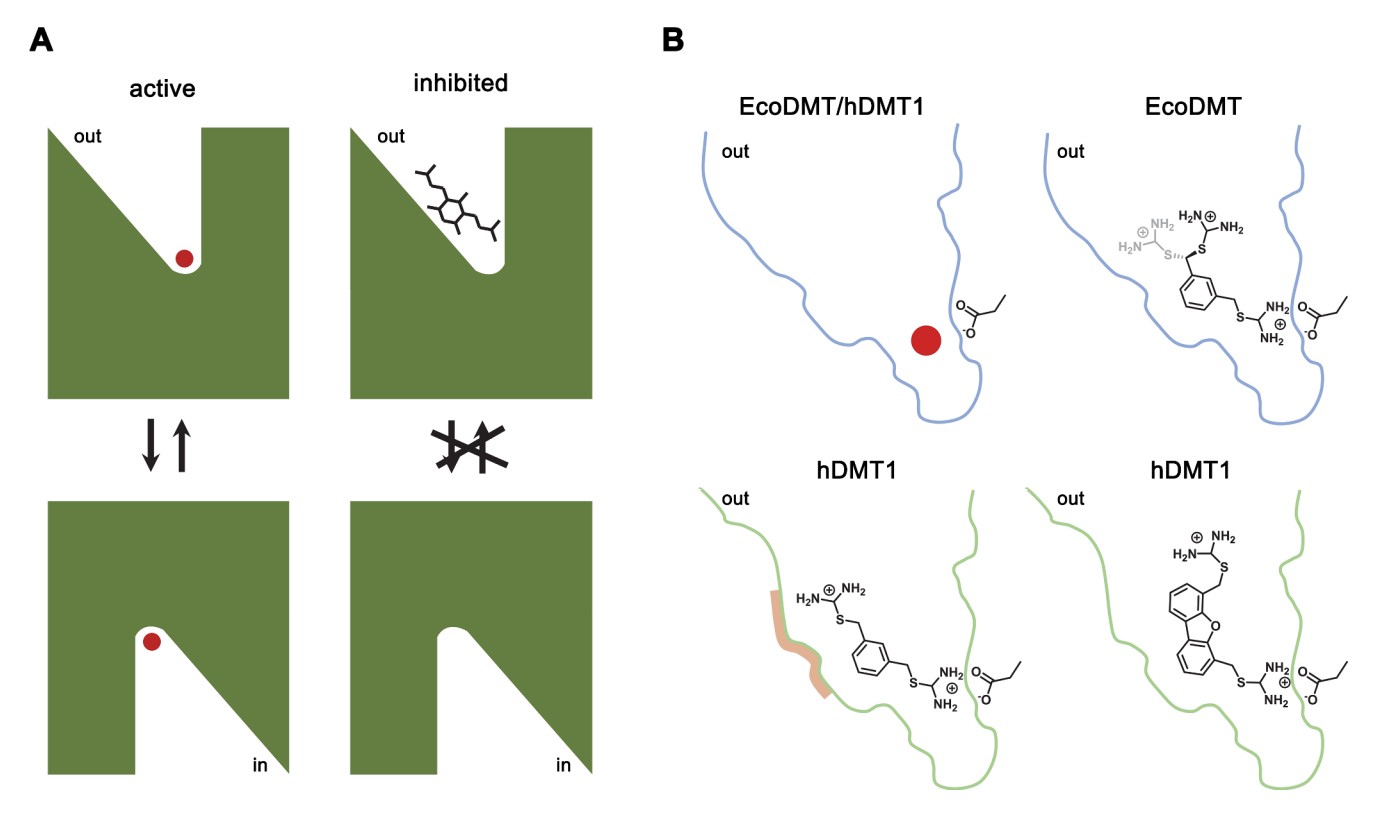

**Figure 6.** Inhibition mechanism. (**A**) Scheme of the inhibitor binding to the outward-facing cavity of an SLC11 transporter thereby preventing substrate binding and conformational changes. (**B**) Interactions of the transported ion and different inhibitors with the outward-facing cavity of EcoDMT and hDMT1. The shape of the cavity is indicated, the sidechain of the conserved aspartate of the metal ion binding site and the chemical structures of different inhibitors are shown.

shaped pocket is sufficiently wide to accommodate larger groups, which might undergo successively stronger interactions, which is illustrated by the increased potency of two compounds containing additional alkyl modification at the aromatic ring (as it is the case for TMBIT and TEBIT) (*Figures 1B* and *6B*; *Figure 1—figure supplement 2B,C*). This general mode of interaction might also explain the inhibition of isothiourea-based compounds with larger ring systems as it is the case for the dibenzofuran Br-DBFIT (*Figures 1B* and *6B*; *Figure 1—figure supplement 2A*) and related compounds characterized in a previous study (*Zhang et al., 2012*). Whereas one isothiourea group strongly interacts with the metal ion binding site in both pro- and eukaryotic transporters, the opposite groups reside in the wider exit of the cavity in a region that is poorly conserved between different SLC11 homologues (*Figure 3E,G*, *Figure 3—figure supplement 2A*). In EcoDMT it most likely undergoes no specific interactions with the protein and instead exhibits large conformational flexibility as supported by the absence of electron density for this group in the X-ray structure of the inhibitor complex and by the mostly unaltered potency in mutants of potentially interacting residues (*Figures 3A,B*, *4* and *6B*). In contrast, mutations of equivalent positions in human DMT1 show a more pronounced effect thus pointing towards stronger inhibitor interactions distal to the metal binding site compared to EcoDMT (*Figures 5* and *6B*). This is generally supported by the reduced potency of an asymmetric compound binding to human DMT1 where only one of the isothiourea groups was modified (*Figure 1—figure supplement 2F*). In this case the effect of the modification could be explained by a moderate decrease in the interaction at the distal side where interactions with the protein might be less specific and by the reduced entropy of binding of the asymmetric compound with the metal ion binding site, which demands interaction with the isothiourea group (*Figure 6*). A strategy to increase the potency and selectivity of compounds towards human DMT1

could thus rely on the optimization of interactions at the distal side of the binding pocket by a systematic variation of aromatic scaffolds and attached polar groups.

In summary we have provided the first detailed mechanistic insight into the pharmacology of transition metal transporters of the SLC11 family. Our results are relevant for potential therapeutic strategies inhibiting human DMT1, which could be beneficial in cases where excessive uptake of iron in the intestine leads to iron overload disorders as observed in hereditary or secondary hemochromatosis, and our study provides a framework that might aid the improvement of these compounds to optimize both their potency and specificity.

# Materials and methods

## Key resources table

| Reagent type (species) or resource | Designation | Source or reference | Identifier | Additional information |
|---|---|---|---|---|
| Chemical compound, drug | Opti-MEM | Thermo fisher Scientific | Cat#11058021 | |
| Chemical compound, drug | Lipofectamine 2000 Transfection Reagent | Thermo fisher Scientific | Cat#11668027 | |
| Chemical compound, drug | Dulbecco's Modified Eagle's Medium - high glucose | MERCK | Cat#D6429 | |
| Chemical compound, drug | Fetal Bovine Serum (FBS) | MERCK | Cat#F7524 | |
| Chemical compound, drug | MEM Non-essential Amino Acid Solution (100x) | MERCK | Cat#M7145 | |
| Chemical compound, drug | Poly-D-lysine hydrobromide | MERCK | Cat#P6407 | |
| Chemical compound, drug | Penicillin-Streptomycin | MERCK | Cat#P4333 | |
| Chemical compound, drug | HEPES solution | MERCK | Cat#H0887 | |
| Chemical compound, drug | Sodium pyruvate solution | MERCK | Cat# S8636 | |
| Chemical compound, drug | MicroScint-20 | PerkinElmer | Cat# 6013621 | |
| Chemical compound, drug | Iron-55 Radionuclide as Ferric chloride/s.A.>3 Ci/g, Ferric chloride in 0.5MHCl | ANAWA Biomedical Services and Products | Cat# ARX-0109–2 | |
| Chemical compound, drug | Terrific Broth (TB) medium | Sigma | Cat#T9179 | |
| Chemical compound, drug | Ampicillin | AppliChem | Cat#A0839 | |
| Chemical compound, drug | L-(+)-Arabinose | Sigma-Aldrich | Cat#A3256 | |

*Continued on next page*

*Continued*

| Reagent type (species) or resource | Designation | Source or reference | Identifier | Additional information |
|---|---|---|---|---|
| Chemical compound, drug | Lysozyme | AppliChem | Cat#A3711 | |
| Chemical compound, drug | DNase I | AppliChem | Cat#A3778 | |
| Chemical compound, drug | n-decyl-β-D-maltoside (DM) | Anatrace | Cat#D322 | |
| Chemical compound, drug | n-dodecyl-β-D-maltoside (DDM) | Anatrace | Cat#D310 | |
| Chemical compound, drug | 1-palmitoyl-2-oleoyl-sn-glycero-3-phosphoethanolamine (POPE) | Avanti Polar Lipids, Inc | Cat#850757 | |
| Chemical compound, drug | 1-palmitoyl-2-oleoyl-sn-glycero-3-phospho-(1'-rac-glycerol) (POPG) | Avanti Polar Lipids, Inc | Cat#840457 | |
| Chemical compound, drug | Diethyl ether | Millipore Sigma | Cat#296082 | |
| Chemical compound, drug | Triton X-100 | Millipore Sigma | Cat#T9284 | |
| Chemical compound, drug | Calcein | Thermo Fisher Scientific | Cat#C481 | |
| Chemical compound, drug | Valinomycin | Thermo Fisher Scientific | Cat#V1644 | |
| Chemical compound, drug | A-23187 Free Acid (Calcimycin) | Thermo Fisher Scientific | Cat#A1493 | |
| Peptide, recombinant protein | HRV 3C protease | Expressed and purified (Expression vector pET-3C) Raimund Dutzler laboratory | N/A | |
| Commercial assay or kit | Corning96 Well Black Polystyrene Microplate | MERCK | Cat# CLS3603 | |
| Commercial assay or kit | PfuUltra High-fidelity DNA Polymerase | Agilent Technologies | Cat# 600380 | |
| Commercial assay or kit | Nickel NTA Agarose Resin | ABT Agarose Bead Technologies | 6BCL-NTANi-X | |
| Commercial assay or kit | Superdex 200 10/300 GL | GE Healthcare | Cat#17517501 | |
| Commercial assay or kit | 24-well Cryschem M crystallization plate | Hampton Research | Cat#HR1-002 | |
| Commercial assay or kit | Bio-Beads SM-2 | Bio-Rad | Cat#1523920 | |
| Commercial assay or kit | Avestin LiposoFast Liposome Factory Basic | Millipore Sigma | Cat#Z373400 | |

*Continued on next page*

*Continued*

| Reagent type (species) or resource | Designation | Source or reference | Identifier | Additional information |
|---|---|---|---|---|
| Commercial assay or kit | 400 nm polycarbonate filters for LiposoFast | Millipore Sigma | Cat#Z373435 | |
| Commercial assay or kit | 96-well black-walled microplate | Thermo Fisher Scientific | Cat#M33089 | |
| Commercial assay or kit | Amicon 50 kDa MWCO centrifugal filter | EMD Millipore | Cat#UFC805024 | |
| Commercial assay or kit | 0.22 µm Ultrafree-MC Centrifugal Filter | EMD Millipore | Cat#UFC30GV | |
| Strain, strain background | *Escherichia coli* MC1061 | Thermo Fisher Scientific | Cat#C66303 | |
| Strain, strain background | *Eremococcus coleocola* strain | https://www.dsmz.de/collection | DSM No. 15696 | |
| Cell line (human) | HEK-293 | ATCC | CRL-1573 | |
| Cell line (human) | HEK-293T | ATCC | CRL-3216 | |
| Recombinant DNA | Human SLC11A2 isoform 1A-IRE (+) ORF inserted in pBluescript SK - vector | Hentze laboratory- EMBL, Heildelberg, Germnay | N/A | |
| Recombinant DNA | pIRES2 DsRed-Express2 Vector | Takara Clontech | Cat# 632540 | |
| Recombinant DNA | Expression vector pBXC3GH | Addgene | Cat#47070 | |
| Recombinant DNA | Expression vector pBXC3H | Addgene | Cat#47068 | |
| Recombinant DNA | Expression vector pET-3C | Dr. Arie Geerlof, EMBL Munich | N/A | |
| Software, algorithm | XDS | *Kabsch, 2010* | http://xds.mpimf-heidelberg.mpg.de/ | |
| Software, algorithm | CCP4 | *Collaborative Computational Project, 1994* | http://www.ccp4.ac.uk/ | |
| Software, algorithm | Coot 0.8.9 | *Emsley and Cowtan, 2004* | https://www2.mrc-lmb.cam.ac.uk/personal/pemsley/coot/ | |
| Software, algorithm | PHENIX 1.17 | *Adams et al., 2002* | http://phenix-online.org/ | |
| Software, algorithm | MSMS | *Sanner et al., 1996* | http://mgltools.scripps.edu/packages/MSMS/ | |
| Software, algorithm | DINO 0.9.4 | http://www.dino3d.org | http://www.dino3d.org | |
| Software, algorithm | LIGPLOT | *Wallace et al., 1995* | https://www.ebi.ac.uk/thornton-srv/software/LIGPLOT/ | |
| Software, algorithm | CHARMM | *Brooks et al., 1983* | https://www.charmm.org/charmm/ | |
| Software, algorithm | SWISS-MODEL | *Biasini et al., 2014* | https://swissmodel.expasy.org/ | |
| Software, algorithm | Prism 8.3.0 | GraphPad | https://www.graphpad.com/ | |

*Continued on next page*

*Continued*

| Reagent type (species) or resource | Designation | Source or reference | Identifier | Additional information |
|---|---|---|---|---|
| Software, algorithm | Microcal Origin 5.0 | Microcal Software, Inc. | http://www.microcal.com | |
| Software, algorithm | MicroCal Concat ITC | Malvern Panalytical | https://www.malvernpanalytical.com/en/ | |

### Chemical synthesis

The chemical synthesis of all compounds is described in Appendix 1.

### Cell lines

Experiments using human cell lines were conducted with HEK293 cells either stably (ATCC-CRL-1573) or transiently (ATCC-CRL3216) over-expressing DsRED-hDMT1 constructs. The cell line stably over-expressing hDMT1 has been characterized previously (*Montalbetti et al., 2014*). Mycoplasma contamination was negative for both cell lines as tested with the LooKOut Mycoplasm PCR Detection Kit (Sigma-MP0035). All cells were grown in DMEM media (Invitrogen) supplemented with 10% FBS, 10 mM HEPES and 1 mM Na-pyruvate at 37°C, 95% humidity and air containing 5% $CO_2$. For cells stably over-expressing hDMT1, the media was additionally supplemented with 500 μg ml$^{-1}$ geneticin (Life Technologies).

### Construct generation

The sequence coding for the hDMT1 isoform 1A-IRE (+) (UniProt identifier P49281-3) was cloned into pIRES2 DsRed-Express2 bicistronic vector (*Montalbetti et al., 2014*). Single-point mutations A291G, A291V, Q475A, Q475F, S476V, L479F, N520L, F523A and Y527A were introduced into the hDMT1 encoding sequence as previously described (*Pujol-Giménez et al., 2017*). For EcoDMT, the corresponding gene (UniProtKB identifier E4KPW4) was cloned using genomic DNA isolated from a *Eremococcus coleocola* strain (DSM No. 15696) into the arabinose-inducible expression vectors pBXC3GH and pBXC3H with fragment-exchange (FX) cloning (*Geertsma and Dutzler, 2011*). The point mutations N456A, N456L, S459A and Q463A were introduced by site-directed mutagenesis (*Li et al., 2008*).

### Iron uptake and inhibition assays for hDMT1

For uptake experiments, HEK293 cells were grown in clear bottom, white-well, poly-D-lysine coated 96 well plates (Corning). Cells stably over-expressing hDMT1 were seeded 24 hr before the experiment at a density of 50,000 cells/well and cells used for transient transfection were seeded at 30.000 cells/well for 48 hr prior to the experiment and transfected 24 hr before the experiment using Lipofectamine 2000 (Life technologies) as described in the manufacturer's protocol. Briefly, culture media was removed from the wells and the cells were washed three times with uptake buffer (140 mM NaCl, 2.5 mM KCl, 1 mM CaCl$_2$, 1 mM MgCl$_2$, 1.2 mM K$_2$HPO$_4$, 10 mM glucose, 5 mM HEPES, 5 mM MES, pH 7.4). After the wash, the cells were incubated for 15 min at room temperature (RT) with uptake solution containing the indicated amount of non-radioactive ferrous iron (Fe$^{2+}$), 100 μM Ascorbic acid and 0.5 μCi/ml radiolabeled $^{55}$Fe$^{2+}$ (American Radiolabeled) dissolved in uptake buffer (pH 5.5). After incubation, uptake solution was removed from the wells, and the cells were washed three times in ice-cold uptake buffer (pH 7.5). Before quantification, a scintillation cocktail (Mycrosinth 20, PerkinElmer) was added to each well, and the cells were incubated during 1 hr at RT under constant agitation. Accumulated radioactivity was measured using a TopCount Microplate Scintillation Counter (PerkinElmer). Transport rates were quantified with:

$$influx\ rate = \frac{counts/well\ (cpm) \times [substrate]\ (pM)}{total\ counts\ (cpm/L) \times uptake\ time\ (\min)} \tag{1}$$

To assess their inhibitory effect, cells were incubated with the indicated compounds at the specified concentrations during 5 min at RT prior to the addition of the uptake solution. To determine the

kinetic parameters for the $Fe^{2+}$ transport mediated by hDMT1 WT and point mutants, the influx rates at different iron concentrations were fitted to the Michaelis-Menten equation. For the determination of $IC_{50}$ values, influx rates for each inhibitor concentration were plotted and data was fitted to a 4-parameter sigmoidal curve. Plotted influx rates correspond to the mean of the indicated biological replicates, errors are s.d. Each experiment was performed in duplicates for transiently transfected cells with data obtained from at least two independent transfections or triplicates for stably overexpressed WT hDMT1.

## Expression and purification of EcoDMT

EcoDMT WT and mutants were expressed in *E. coli* MC1061 as C-terminally-tagged fusion proteins containing a 3C-protease cleavage site followed by a $His_{10}$-tag. The tag was removed during purification unless specified otherwise. *E. coli* cells were grown in Terrific Broth (TB) medium supplemented with 100 µg $ml^{-1}$ ampicillin, either by fermentation or in flasks. Cells were grown at 37°C and the temperature was gradually decreased to 25°C prior to induction. Protein expression was induced by addition of 0.0045% (w/v) L-arabinose at an $OD_{600}$ of ~2.5 for fermenter cultures and ~0.8 for cultures in flasks. For overnight expression the temperature was decreased to 18°C and cells were subsequently harvested by centrifugation. All following protein purification steps were carried out at 4° C. The cells were lysed in buffer A (20 mM HEPES, pH 7.5, and 150 mM NaCl) supplemented with 1 mg $ml^{-1}$ (w/v) lysozyme and 20 µg $ml^{-1}$ DNaseI using HPL6 high-pressure cell disruptor (MAXIMATOR). The lysate was subjected to a low-spin centrifugation (10,000 g for 20 min) and subsequently the membrane vesicles were harvested by ultracentrifugation (200,000 g for 1 hr). Membrane proteins were extracted by resuspending the vesicles in buffer A containing 10% (w/v) glycerol and 1–2% (w/v) of the specified detergents and subsequently the extract was cleared by centrifugation. The detergent *n*-decyl-β-D-maltopyranoside (DM, Anatrace) was used to purify proteins for reconstitution or crystallization experiments and *n*-dodecyl-β-D-maltopyranoside (DDM, Anatrace) for isothermal titration calorimetry (ITC). The extracted proteins were purified by immobilized metal affinity chromatography (IMAC). The GFP- $His_{10}$ tag was removed by addition of HRV-3C protease at a protein:protease molar ratio of 5:1 for 2 hr while dialyzing the sample against 20 mM HEPES, pH 7.5, 150 mM NaCl, 8.7% (w/v) glycerol, and 0.1% (w/v) DM or 0.04% (w/v) DDM. A second IMAC step was used to separate the GFP-$His_{10}$ tag and the protease from the cleaved protein. Subsequently, the purified membrane proteins were subjected to size exclusion chromatography on a Superdex S200 column (GE Healthcare) equilibrated in 10 to 20 mM HEPES, pH 7.5, 150 mM NaCl, and either 0.25% (w/v) DM or 0.04% (w/v) DDM. Peak fractions were used for reconstitution into liposomes, ITC and crystallization experiments. Purified samples of WT and mutant proteins were analyzed by SDS-PAGE and mass spectrometry.

## X-ray structure determination

Crystals of EcoDMT were grown in 24-well plates in sitting drops at 4°C by mixing 1 µl of protein (at a concentration of 7–10 mg $ml^{-1}$) with 1 µl of reservoir solution consisting of 50 mM Tris-HCl pH 8.0–9.0 and 22–26% PEG 400 (v/v) and equilibrated against 500 µl of reservoir solution. Crystals grew within two weeks. For preparation of inhibitor complexes, crystals were soaked for several minutes with either Br-BIT or oBr-BIT. The two inhibitors were either added to the cryoprotection solutions at a final concentration of 5 mM or directly added as powder to the drops containing the crystals. For cryoprotection, the PEG 400 concentration was increased stepwise to 35% (v/v). All data sets were collected on frozen crystals on the X06SA or the X06DA beamline at the Swiss Light Source of the Paul Scherrer Institute on an EIGER X 16M or a PILATUS 6M detector (Dectris). Anomalous data were collected at the bromine absorption edge (0.92 Å). Data were integrated and scaled with XDS (*Kabsch, 2010*) and further processed with CCP4 programs (*Collaborative Computational Project, 1994*). Structures were refined in Phenix (*Adams et al., 2002*) using the EcoDMT WT structure (PDB ID 5M87) as starting model. The model was modified in COOT (*Emsley and Cowtan, 2004*) and constraints for the refinement of the Br-BIT ligand were generated using the CCP4 program PRODRG (*Schüttelkopf and van Aalten, 2004*). Five percent of the reflections not used in refinement were used to calculate $R_{free}$. The final refinement statistics is reported in *Table 2*. The coordinates of the EcoDMT-Br-BIT complex refined to data at 3.8 Å were deposited with the PDB under accession code 6TL2.

## Modeling and Poisson-Boltzmann calculations

The electrostatic potential in the extracellular aqueous cavity harboring the inhibitor binding site was calculated by solving the linearized Poisson–Boltzmann equation in CHARMM (*Brooks et al., 1983*; *Im et al., 1998*) on a 150 Å ×150 Å × 200 Å grid (1 Å grid spacing) followed by focusing on a 100 Å x 100 Å x 120 Å grid (0.5 Å grid spacing). Partial protein charges were derived from the CHARMM36 all-hydrogen atom force field. Hydrogen positions were generated in CHARMM, histidines were protonated. The protein was assigned a dielectric constant ($\epsilon$) of 2. Its transmembrane region was embedded in a 30 Å-thick slab ($\epsilon = 2$) representing the hydrophobic core of the membrane and two adjacent 10 Å-thick regions ($\epsilon = 30$) representing the headgroups. The membrane region contained a 38 Å-high and 22 Å-wide aqueous cylinder ($\epsilon = 80$) covering the extracellular aqueous cavity and was surrounded by an aqueous environment ($\epsilon = 80$). Calculations were carried out in the absence of monovalent mobile ions in the aqueous regions. The homology model of human DMT1 was prepared with the SWISS-MODEL homology modeling server (*Biasini et al., 2014*).

## Reconstitution of EcoDMT into liposomes

EcoDMT WT and mutants were reconstituted using detergent destabilized liposomes according to *Geertsma et al. (2008)*. The liposomes were formed using the synthetic phospholipids POPE and POPG (Avanti Polar lipids) at a w/w ratio of 3:1. The lipids where resuspended in 20 mM HEPES, pH 7.5, and 100 mM KCl after washing with diethylether and drying by exsiccation. Liposomes were subjected to three freeze-thaw cycles and extruded through a 400 nm polycarbonate filter (Avestin, LiposoFast-Basic) to form unilammellar vesicles. Triton X-100 was used to destabilize the liposomes and the reconstitutions were performed at a protein to lipid ratio of 1:100 (w/w) for transport assays and a protein to lipid ratio of 1:50 (w/w) to determine the orientation of the transporters in the liposomes. After detergent removal by the successive addition of Bio-Beads SM-2 (Bio-Rad) over a period of three days, proteoliposomes were harvested by centrifugation, resuspended in buffer containing 20 mM HEPES, pH 7.5, and 100 mM KCl and stored in liquid nitrogen.

The orientation of the transporters in proteoliposomes was determined using a reconstitution of EcoDMT-His$_{10}$ in which the C-terminally Histidine-tag preceded by a 3C protease cleavage site has not been cleaved prior to reconstitution. Initially, proteoliposomes (containing a total of 2 mg lipids) were extruded using a 400 nm polycarbonate filter to generate unilammellar vesicles and split in two equal aliquots. Purified 3C protease was subsequently added to the outside of one aliquot of the proteoliposmes and incubated for 2 hr at room temperature. The external 3C protease was removed by washing twice with 20 volumes of 20 mM HEPES, pH 7.5, and 100 mM KCl and the liposomes were harvested by centrifugation. After removal of the protease, the liposomes were dissolved by addition of DM at a detergent to lipid ration of 1.25:1 (w/w) with half of the samples incubated with 3C protease for 2 hr on ice. All 3C cleavage steps were performed with a large excess of protease to ensure completion of the reaction. Control liposomes not treated with 3C protease at the different steps were processed the same way. A sample of purified EcoDMT-His$_{10}$ was used as control to follow the removal of the His$_{10}$-tag in a sample with unrestricted accessibility to the 3C cleavage site. The final samples were analyzed by SDS-PAGE.

## Fluorescence-based Mn$^{2+}$ transport and inhibition assays

Proteoliposomes for the Mn$^{2+}$ transport and inhibition assays were obtained by resuspension of vesicles in buffer B containing 20 mM HEPES, pH 7.5, 100 mM KCl and 250 µM calcein (Invitrogen) and subjection to three freeze-thaw cycles followed by extrusion through a 400 nm filter. Proteoliposomes were harvested by centrifugation and washed twice with 20 volumes of buffer B without Calcein. The samples were subsequently diluted to 0.25 mg lipid ml$^{-1}$ in buffer containing 20 mM HEPES, pH 7.5 and 100 mM NaCl and varying concentrations of TMBIT, Br-BIT or oBr-BIT. Subsequently, 100 µl aliquots were placed in a black 96-well plate and after stabilization of the fluorescence signal, valinomycin (at a final concentration of 100 nM) and MnCl$_2$ were added to start the assay. Uptake of Mn$^{2+}$ into liposomes was recorded by measuring the fluorescence change in a fluorimeter (Tecan Infinite M1000, $\lambda_{ex}$=492 nm/ $\lambda_{em}$=518 nm) in four-second intervals. As a positive control, Mn$^{2+}$ ions were equilibrated by addition of the ionophore calcimycin (at a final concentration of 100 nM) (Invitrogen), which acts as a Mn$^{2+}$/H$^+$ exchanger at the end of the experiments. In presence of TMBIT or oBr-BIT at concentrations higher than 50 µM, the fluorescence signal after addition of

calcimycin did not reach the same low level as observed in absence of inhibitors, which suggests an interference of the compounds with the activity of calcimycin at high concentrations. Initial transport rates ($\Delta F\ min^{-1}$) were obtained by performing a linear regression of transport data obtained between 60 and 120 s after addition of valinomycin and $MnCl_2$ and fitted to a Michaelis-Menten equation. Kinetic data of WT and all mutants described in this study was measured in at least three independent experiments.

## Analysis of kinetic data

Kinetic data was fitted to a mixed enzyme inhibition model outlined below (*Scheme 1*) (*Copeland, 2005*) with GraphPad Prism (GraphPad Software, San Diego, California USA, www.graphpad.com):

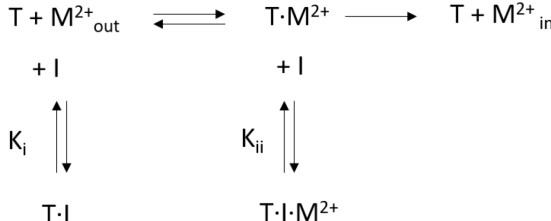

**Scheme 1.** Mixed enzyme inhibition model.

This model assumes that the inhibitor (I) binds to the substrate free transporter (T) and to the transporter-substrate complex ($T \cdot M^{2+}$) with equilibrium constants $K_i$ and $K_{ii}$, respectively. Both equilibrium constants can be obtained by a non-linear regression to *Equation 2*

$$v = \frac{Vmax \cdot S}{Km \cdot \left(1 + \frac{I}{Ki}\right) + S \cdot \left(1 + \frac{I}{Kii}\right)} \ \text{with} \ Kii = \alpha \cdot Ki \tag{2}$$

For high values of $\alpha$, the inhibitor preferentially binds to the substrate-free transporter and *Equation 2* approaches a model for competitive inhibition. The resulting equilibrium constants obtained for hDMT1 using a radioactive $^{55}Fe^{2+}$ transport assay and for EcoDMT1 using an in vitro proteoliposome-based assay are summarized in *Table 1*.

## Isothermal titration calorimetry

Isothermal titration calorimetry experiments were performed with a MicroCal ITC200 system (GE Healthcare). The titrations of $MnCl_2$ to TMBIT and Br-BIT were performed at 25°C in 20 mM HEPES, pH 7.5, and 100 mM KCl. The syringe was filled with 5 mM $MnCl_2$ and sequential aliquots of 2 µl were added to the sample cell filled with 0.4 mM TMBIT, Br-BIT, Ethylenediaminetetraacetic acid (EDTA) or buffer. The titrations of Br-BIT to purified EcoDMT were performed at 6°C in 20 mM HEPES, pH 7.5, 150 mM and 0.04% (w/v) DDM. The syringe was filled with 1.8 mM or 2.5 mM Br-BIT and sequential aliquots of 1.5–2 µl were added to the sample cell filled with ~50 µM or ~180 µM EcoDMT WT, the mutant D51A or buffer. Data were analyzed using the Origin ITC analysis package and the MicroCal ITC program Concat and errors on the reported $K_D$ values represent fitting errors. The data were fit using models assuming one or two sets of binding sites. In case of 2.5 mM Br-BIT in the syringe and ~180 µM EcoDMT in the cell, mainly the high affinity step saturating in the low micromolar range is titrated. Therefore, the low affinity transition can be ignored and the resulting reaction enthalpies were y-translated to zero to enable data analysis using a model assuming a single set of binding sites. For each protein, similar results were obtained for at least two experiments from independent protein preparations.

## Data availability

The coordinates and structure factors of the EcoDMT-Br-BIT complex have been deposited in the Protein Data Bank with the accession code 6TL2.

## Acknowledgements

This research was supported by the Swiss National Science Foundation (SNF) through the National Centre of Competence in Research TransCure and the SNF grant 310030_182272 to Matthias A Hediger. Data collection was performed at the X06SA and X06DA beamlines at the Swiss Light Source of the Paul Scherrer Institute and we thank the beamline staff for support during data collection. All members of the Dutzler lab are acknowledged for help in all stages of the project.

## Additional information

### Funding

| Funder | Grant reference number | Author |
| --- | --- | --- |
| Swiss National Science Foundation | NCCR TransCure | Jean-Louis Reymond Matthias A Hediger Raimund Dutzler |
| Swiss National Science Foundation | 310030_182272 | Matthias A Hediger |

The funders had no role in study design, data collection and interpretation, or the decision to submit the work for publication.

### Author contributions

Cristina Manatschal, Jonai Pujol-Giménez, Conceptualization, Data curation, Formal analysis, Validation, Investigation, Visualization, Methodology; Marion Poirier, Conceptualization, Formal analysis, Validation, Methodology; Jean-Louis Reymond, Conceptualization, Resources, Data curation, Formal analysis, Funding acquisition, Validation, Methodology, Project administration; Matthias A Hediger, Conceptualization, Resources, Supervision, Project administration; Raimund Dutzler, Conceptualization, Supervision, Funding acquisition, Project administration

### Author ORCIDs

Cristina Manatschal ⓘ https://orcid.org/0000-0002-4907-7303
Jonai Pujol-Giménez ⓘ https://orcid.org/0000-0002-9951-1390
Jean-Louis Reymond ⓘ https://orcid.org/0000-0003-2724-2942
Raimund Dutzler ⓘ https://orcid.org/0000-0002-2193-6129

### Decision letter and Author response

Decision letter https://doi.org/10.7554/eLife.51913.sa1
Author response https://doi.org/10.7554/eLife.51913.sa2

## Additional files

### Supplementary files

• Transparent reporting form

### Data availability

Coordinates and structure factors have been deposited with the PDB under Accession Code 6TL2.

The following dataset was generated:

| Author(s) | Year | Dataset title | Dataset URL | Database and Identifier |
| --- | --- | --- | --- | --- |
| Manatschal C, Pujol-Giménez J, Poirier M, Reymond J-L, Hediger MA, Dutzler R | 2019 | Coordinates and structure factors | http://www.rcsb.org/structure/6TL2 | Protein Data Bank, 6TL2 |

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

## Appendix 1

## Supporting Information for Chemical Synthesis

### Synthesis

All reagents for the synthesis of compounds 1–7 (***Appendix 1—table 1***) were purchased from commercial sources depending on availability (Sigma Aldrich, TCI or Key organics) and were used without further purification. Absolute ethanol was from VWR. Hexane, diethyl ether, ethyl acetate, dichloromethane and methanol were purchased from Grogg Chemie and distilled prior utilization. HPLC grade acetonitrile was purchased from Fisher scientific. Flash chromatography purifications were performed with silica Gel 60 (Sigma, 0.040–0.063 nm, 230–400 mesh ASTM). Low resolution mass spectra were obtained by electron spray ionization (ESI-MS) in the positive mode on a Thermo Scientific LCQ Fleet. High resolution mass spectra were obtained by electron spray ionization (HR-ESI-MS) in the positive mode recorded on a Thermo Scientific LTQ Orbitrap XL. $^1$H and $^{13}$C-NMR spectra were measured on a Bruker Avance 300 spectrometer (at 300 MHz and 75 MHz, respectively) or on a Bruker Avance II 400 spectrometer (at 400 MHz and 101 MHz, respectively) (***Appendix 1—figures 1–7***). $^1$H and $^{13}$C chemical shifts are quoted relative to solvent signals, and resonance multiplicities are reported as s (singlet), d (doublet), t (triplet), q (quartet), p (pentet), and m (multiplet); br = broad peak. Compound purities were assessed by analytical reversed phase HPLC (RP-HPLC) at a detection wavelength of 214 nm (***Appendix 1—table 2***). Analytical RP-HPLC was performed on a Dionex Ultimate 3000 RSLC System (DAD-3000 RS Photodiode Array Detector) using a Dionex Acclaim RSLC 120 column (C18, 3.0 × 50 mm, particle size 2.2 μm, 120 Å pore size) and a flow rate of 1.2 ml min$^{-1}$. Data were recorded and processed with Dionex Chromeleon Management System (version 6.8) and Xcalibur (version 2.2, Thermo Scientific). Eluents for analytical RP-HPLC were as follows: (A) milliQ-deionized water containing 0.05% TFA, (D) HPLC-grade acetonitrile/milliQ-deionized water (9:1) containing 0.05% TFA. Conditions for analytical RP-HPLC were as follows: in 2.2 min from 100% A to 100% D, then staying on 100% D (method A) in 7.5 min from 100% A to 100% D, then staying on 100% D (method B), GC analyses were carried out on a *Macherey-Nagel* Optima delta-3–0.25 μm capillary column (20 m, 0.25 mm); carrier gas: He 1.4 ml min$^{-1}$; injector: 220˚C split mode; detector: FID 280˚C, H2 35 ml min$^{-1}$, air 350 ml min$^{-1}$. GC–MS analyses were performed on an apparatus equipped with a quadrupole mass analyzer using electron impact (70 eV) fitted with a *Macherey-Nagel* Optima delta-3–0.25 μm capillary column (20 m, 0.25 mm); carrier gas: He 1.4 ml min$^{-1}$; injector: 220˚C split mode.

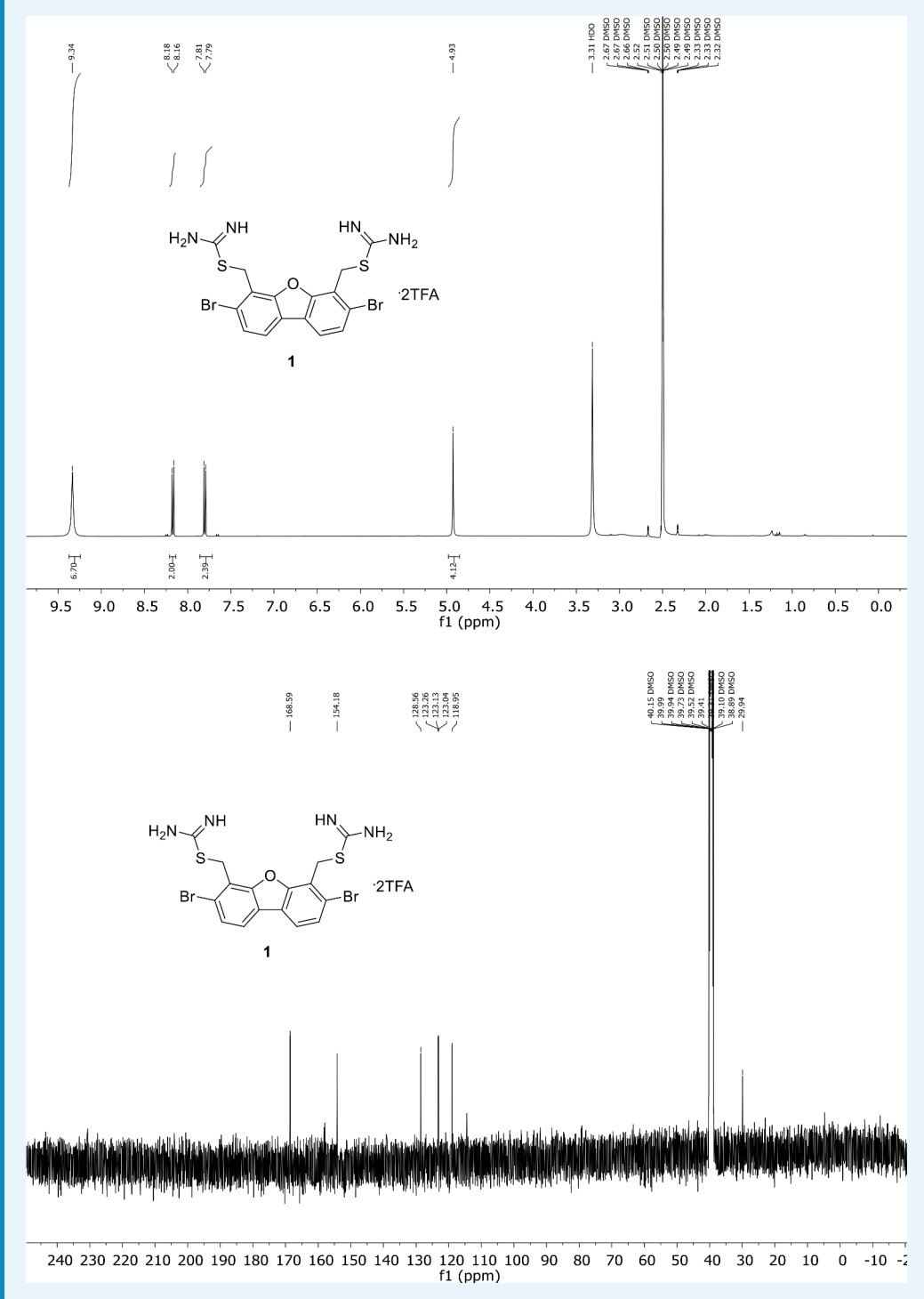

**Appendix 1—figure 1.** Spectroscopic characterization of compound 1. [1]H NMR spectrum of the purified compound is shown on the top, [13]C NMR spectrum on the bottom. Chemical shifts are displayed and listed in the respective synthesis section.

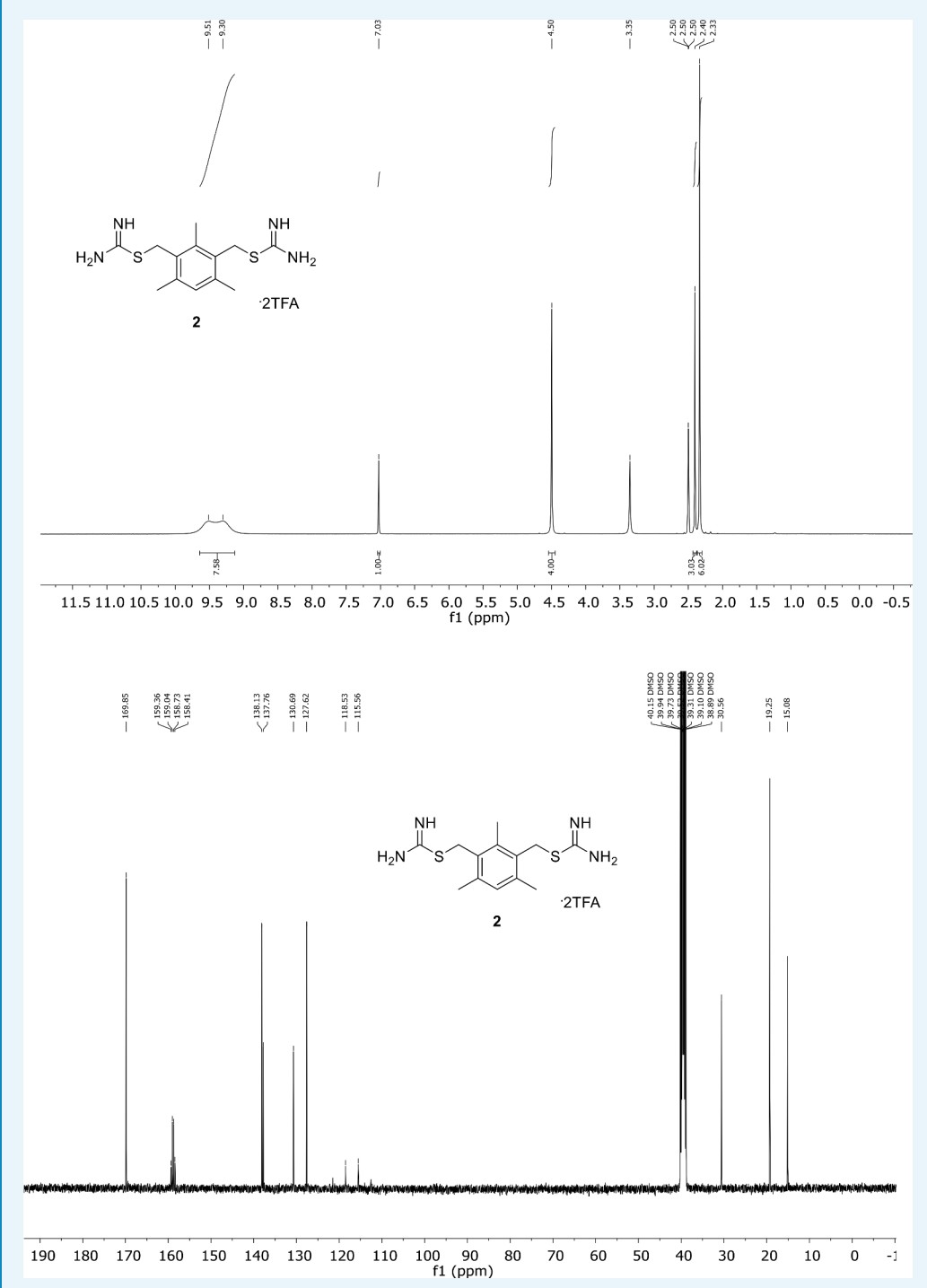

**Appendix 1—figure 2.** Spectroscopic characterization of compound 2. [1]H NMR spectrum of the purified compound is shown on the top, [13]C NMR spectrum on the bottom. Chemical shifts are displayed and listed in the respective synthesis section.

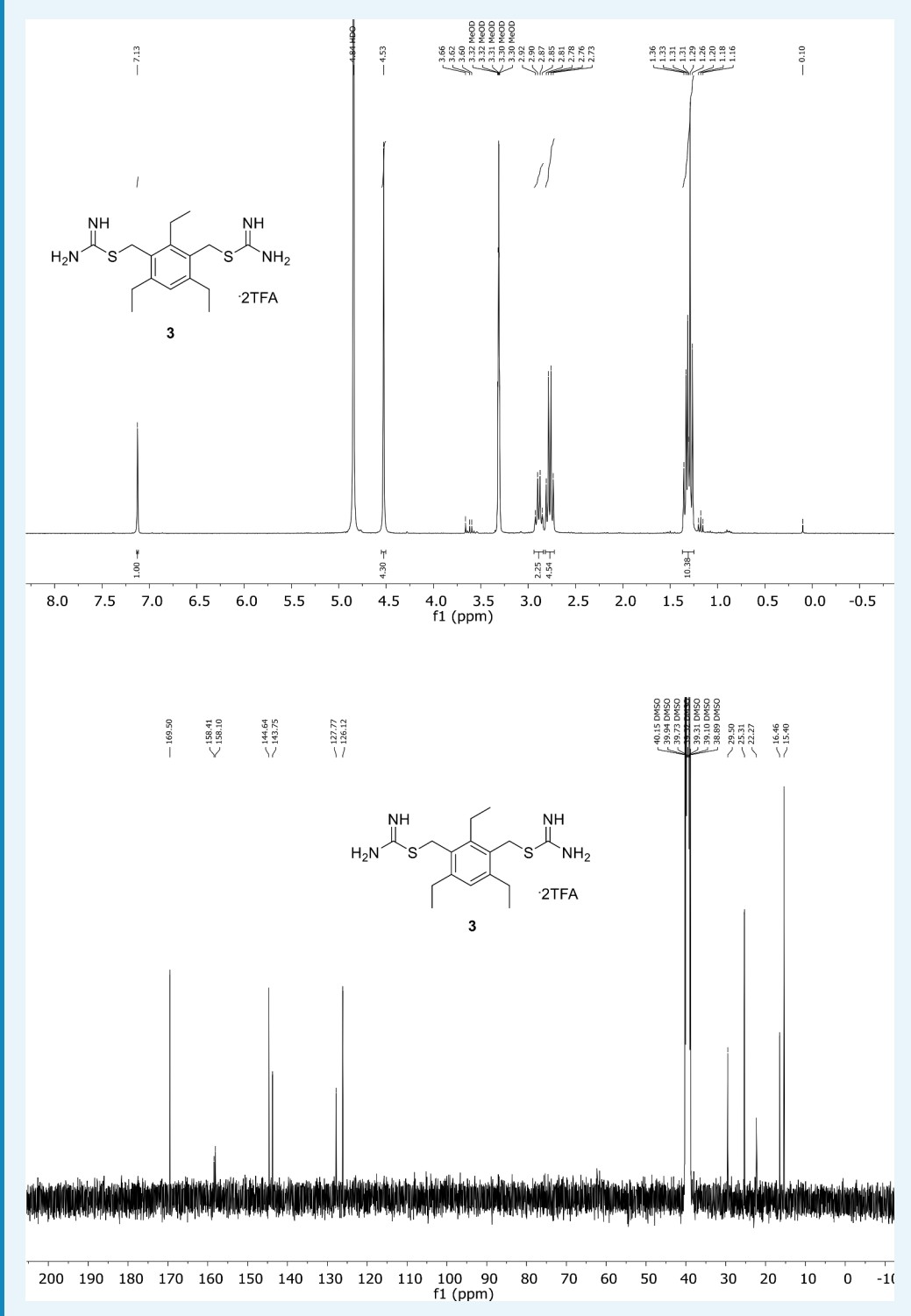

**Appendix 1—figure 3.** Spectroscopic characterization of compound 3. [1]H NMR spectrum of the purified compound is shown on the top, [13]C NMR spectrum on the bottom. Chemical shifts are displayed and listed in the respective synthesis section.

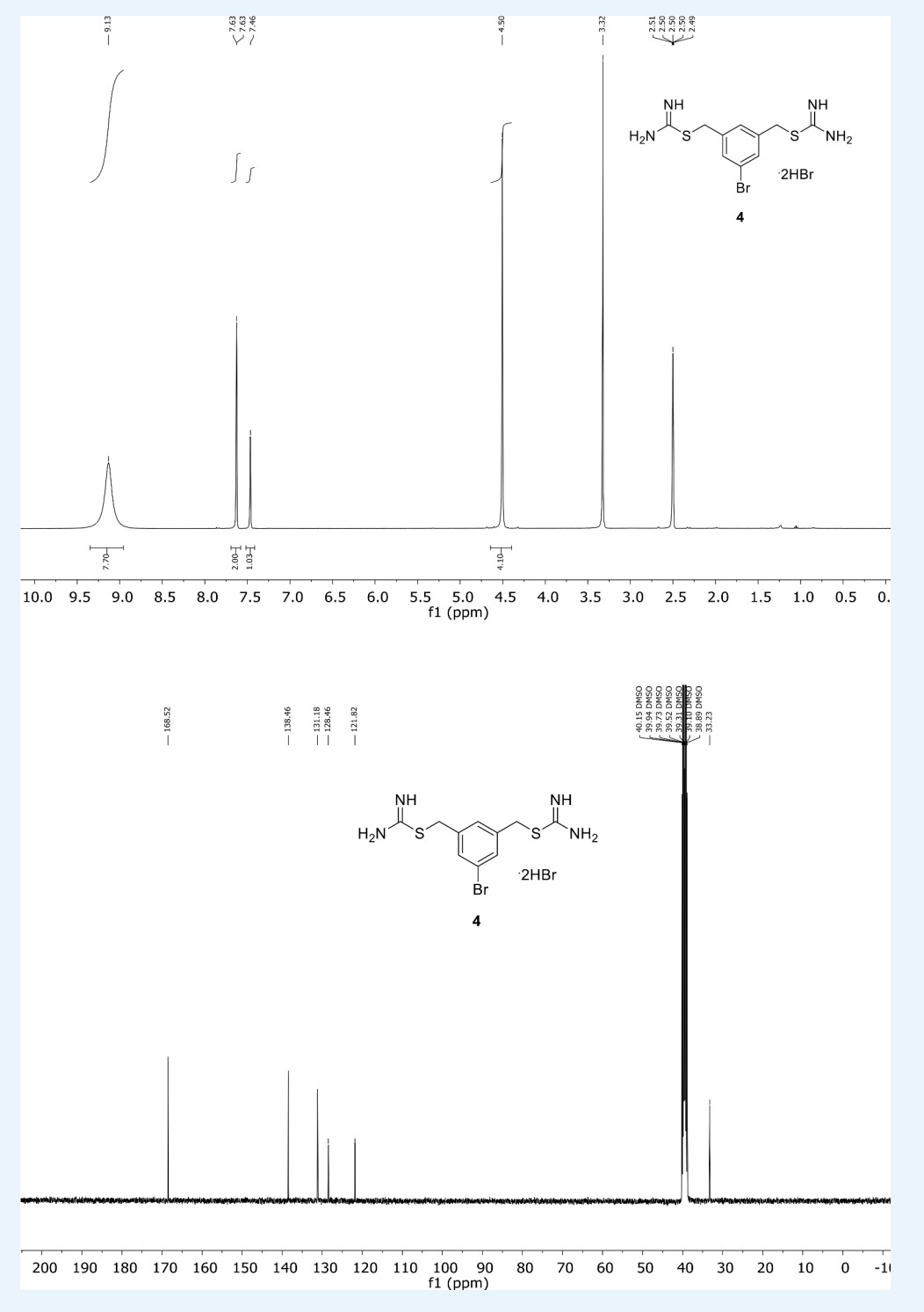

**Appendix 1—figure 4.** Spectroscopic characterization of compound 4. [1]H NMR spectrum of the purified compound is shown on the top, [13]C NMR spectrum on the bottom. Chemical shifts are displayed and listed in the respective synthesis section.

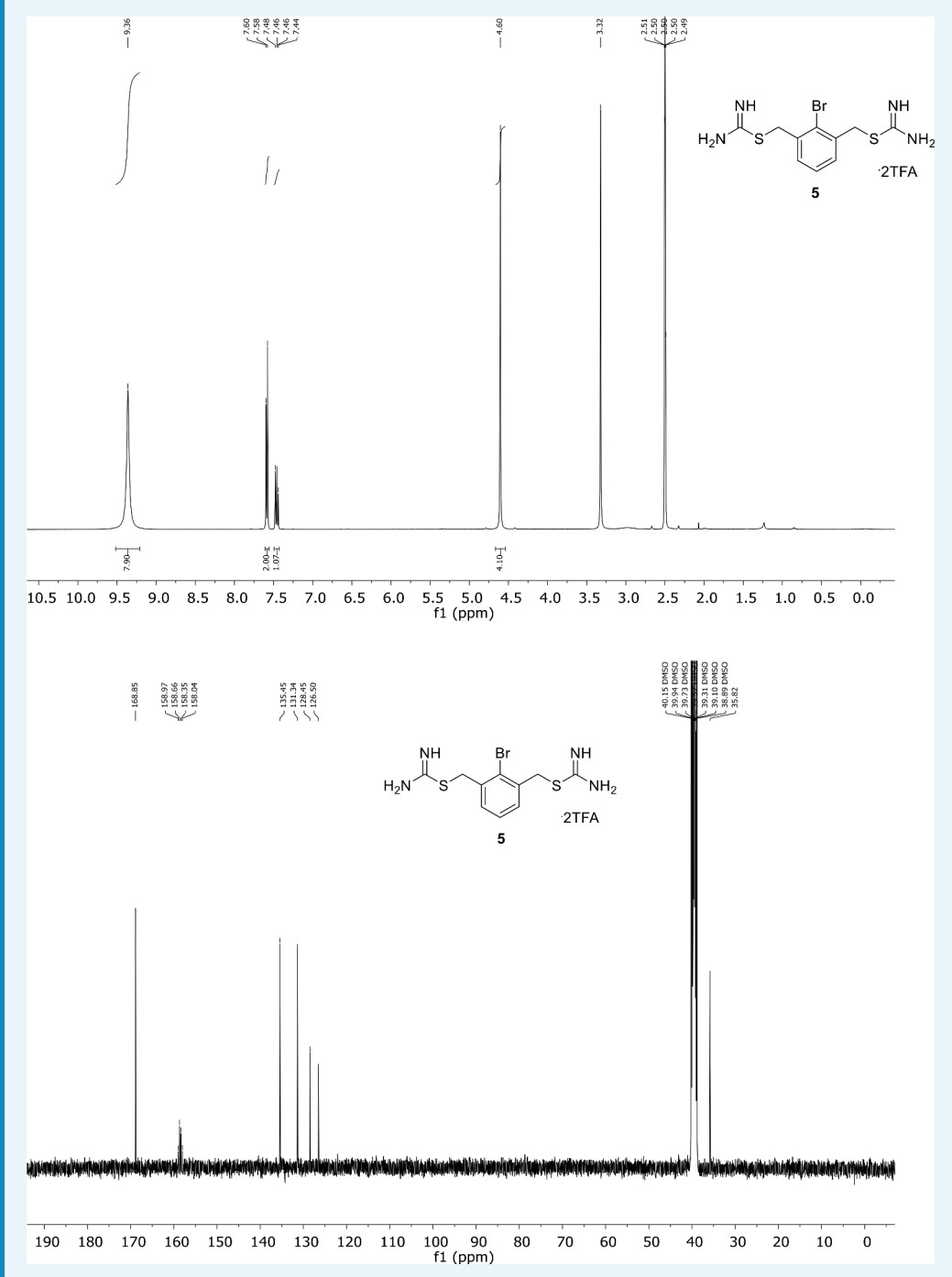

**Appendix 1—figure 5.** Spectroscopic characterization of compound 5. [1]H NMR spectrum of the purified compound is shown on the top, [13]C NMR spectrum on the bottom. Chemical shifts are displayed and listed in the respective synthesis section.

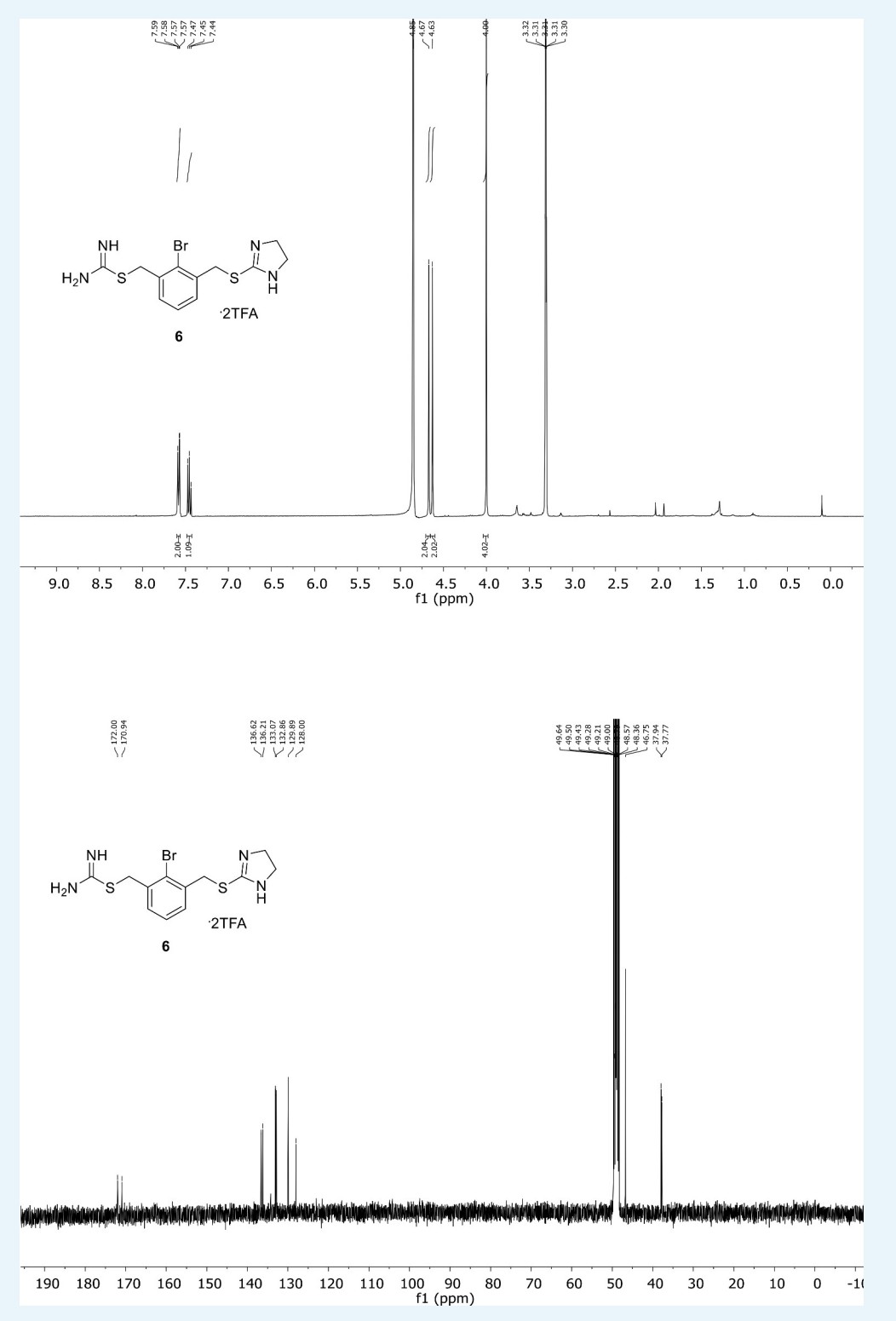

**Appendix 1—figure 6.** Spectroscopic characterization of compound 6. [1]H NMR spectrum of the purified compound is shown on the top, [13]C NMR spectrum on the bottom. Chemical shifts are displayed and listed in the respective synthesis section.

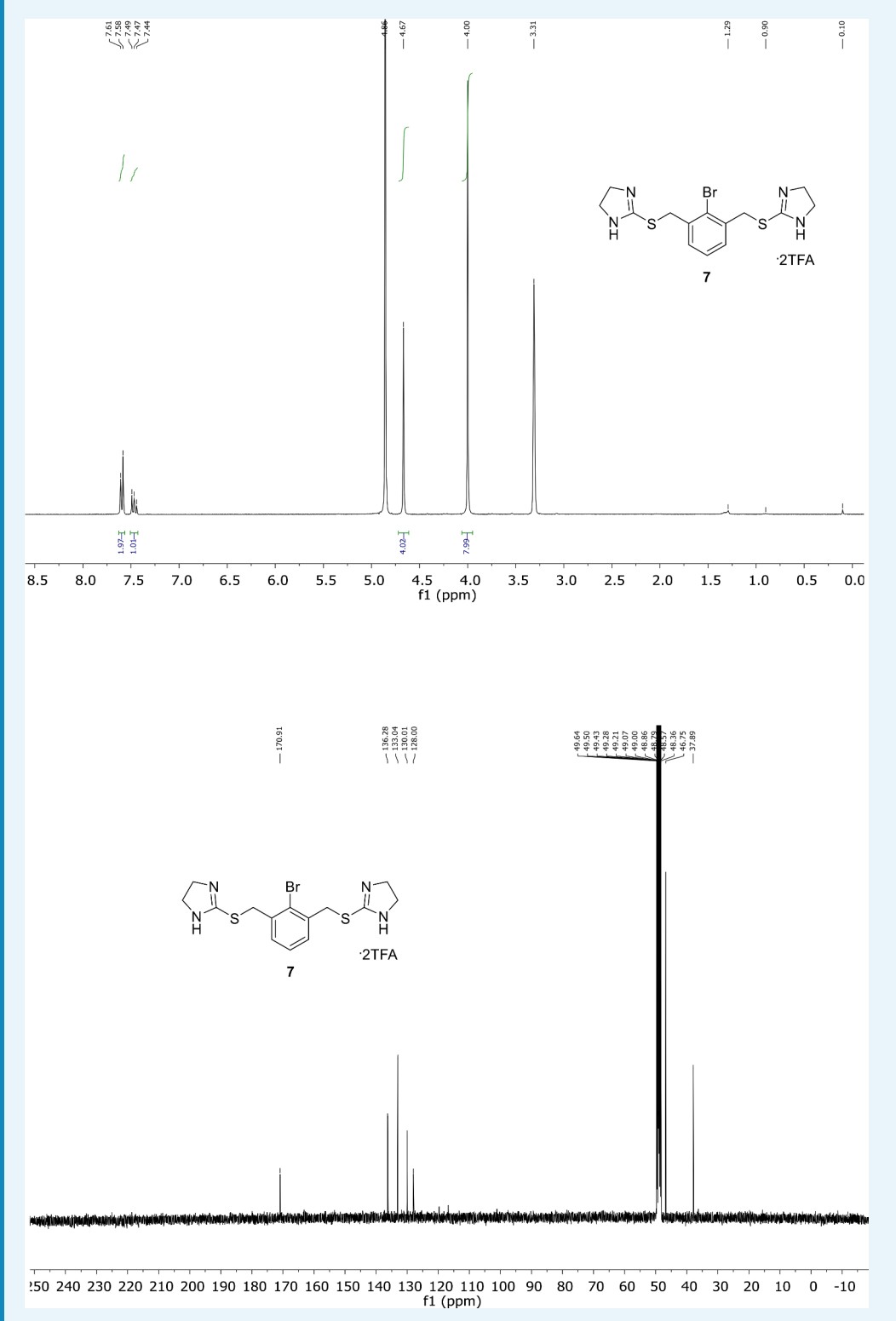

**Appendix 1—figure 7.** Spectroscopic characterization of compound 7. [1]H NMR spectrum of the purified compound is shown on the top, [13]C NMR spectrum on the bottom. Chemical shifts are displayed and listed in the respective synthesis section.

**Appendix 1—table 1.** Compound properties.

| Compound | Smile | IC50 (hDMT1) (μM) | IC50 (hDMT1-S476V) (μM) | IC50 (hDMT1-N520L) (μM) | IC50 (hDMT1-F523A) (μM) |
|---|---|---|---|---|---|
| 1 | BrC1 = CC = C(C(C = CC(Br) =C2CSC(N)=N)=C2O3) C3 = C1CSC(N)=N | 1.24 ± 0.23 | 11.3 ± 0.3 | 2.0 ± 0.4 | 2.9 ± 0.9 |
| 2 | CC1 = C(CSC(N)=N)C(C)=CC(C) =C1CSC(N)=N | 0.35 ± 0.04 | 36.1 ± 8.0 | 21.7 ± 6.9 | 82.9 ± 5.3 |
| 3 | NC(SCC1 = C(CC)C = C(CC)C (CSC(N)=N)=C1 CC)=N | 0.27 ± 0.07 | 32.1 ± 6.3 | 13.7 ± 3.9 | 17.8 ± 7.0 |
| 4 | BrC1 = CC(CSC(N)=N)=CC(CSC (N)=N)=C1 | 4.66 ± 0.4 | 47.6 ± 6.8 | - | 17.3 ± 0.5 |
| 5 | NC(SCC1 = CC = CC(CSC(N) =N)=C1 Br)=N | 2.3 ± 0.25 | | | |
| 6 | NC(SCC1 = CC = CC (CSC2 = NCCN2)=C1 Br)=N | 8.13 ± 0.47 | | | |
| 7 | BrC1 = C(CSC2 = NCCN2) C = CC = C1CSC3 = NCCN3 | 161 ± 24 | | | |

# Definition of method 1

Brominated intermediate (one eq) and thiourea (two eq) were dissolved in ethanol. The solution was stirred at 80˚C overnight and then cooled down to room temperature. Ethanol was removed except 1 ml which was kept. Hexane was added, and the precipitate was filtered and dried *in vacuo* (*Chafeev et al., 2008*).

# Synthesis of (3,7-dibromodibenzo[b,d]furan-4,6-diyl)bis (methylene) dicarbamimidothioate, TFA salt (1 - Br-DBFIT)

Compound **1** was synthesized following method 1 using **10** (169 mg, 0.33 mmol,1.0 eq) and thiourea (57 mg, 0.75 mmol, 2.3 eq) in ethanol (9 ml). The precipitate was washed with diethyl ether/Ethyl acetate (50/50) and then purified by preparative HPLC (gradient from 85/15 A/D to 45/55 A/D in 18 min, $t_R$ = 31% D) to afford **1** (120 mg, 0.33 mmol, 50%) as an off-white solid (*Appendix 1—figure 1*).

RP-UPLC: $t_R$ = 3.27 min (method B). $^1$H NMR (400 MHz, DMSO): δ 9.38 (s, 6H, NH + NH$_2$), 8.17 (2H, d, J = 8.3 Hz), 7.80 (2H, d, J = 8.3 Hz), 4.93 (4H, s, CH$_2$). $^{13}$C NMR (400 MHz, DMSO): δ 168.6, 154.2, 128.6 (CH$_{ar}$), 123.3, 123.2, 123.1 (CH$_{ar}$), 119.0, 29.9 (CH$_2$). HR-MS calculated for C$_{16}$H$_{15}$Br$_2$N$_4$OS$_2$: *m/z* 500.9049, *m/z* found 500.9053.

# Synthesis of (2,4,6-trimethyl-1,3-phenylene)bis (methylene) dicarbamimidothioate, TFA salt (2 -TMBIT)

Compound **2** was synthesized following method 1 using **11** (200 mg, 0.65 mmol, one eq) and thiourea (109 mg, 1.4 mol, 2.2 eq) in ethanol (10 ml). The residue was purified by preparative HPLC (gradient from 95/5 A/D to 75/25 A/D in 18 min, $t_R$ = 16% D) to afford **2** (215 mg, 0.41 mmol, 62%) as a white solid (*Appendix 1—figure 2*).

RP-UPLC: $t_R$ = 1.31 min (method A). $^1$H NMR (400 MHz, DMSO): δ 9.41 (6H, s, NH), 7.03 (1H, s, CH$_{ar}$), 4.50 (4H, s, CH$_2$), 2.40 (3H, s, CH$_3$), 2.33 (6H, s, CH$_3$). $^{13}$C NMR (101 MHz, DMSO): δ 169.9, 158.4, 138.1, 137.8, 130.7 (CH$_{ar}$), 127.6, 30.6 (CH$_2$), 19.3 (CH$_3$), 15.1 (CH$_3$). HR-MS calculated for C$_{13}$H$_{21}$N$_4$S$_2$: *m/z* 297.1202, *m/z* found 287.1205.

## Synthesis of (2,4,6-triethyl-1,3-phenylene)bis(methylene) dicarbamimidothioate, TFA salt (3 - TEBIT)

Compound **3** was synthesized following method 1 using **12** (42 mg, 0.1 mmol, one eq) and thiourea (16 mg, 0.21 mmol, 2.2 eq) in ethanol (4 ml). The residue was purified by preparative HPLC (gradient from 95/5 A/D to 75/25 A/D in 18 min, $t_R$ = 24% D) to afford **3** (52 mg, 0.09 mmol, 94%) as a white solid (**Appendix 1—figure 3**).

RP-UPLC: $t_R$ = 1.52 min (method A). $^1$H NMR (300 MHz, DMSO): δ 9.26 (6H, s, NH + NH$_2$), 7.08 (1H, s, H$_{ar}$), 4.45 (4H, s, CH$_2$), 2.71 (6H, m, C$H_2$CH$_3$), 1.21 (9H, t, $J$ = 7.5 Hz, CH$_3$). $^{13}$C NMR (101 MHz, DMSO): δ 169.5, 144.6, 143.8, 127.8 (CH$_{ar}$), 126.1, 29.5 (CH$_2$), 25.3 (CH$_2$), 22.3 (CH$_2$), 16.5 (CH$_3$), 15.4 (CH$_3$). HR-MS calculated for C$_{16}$H$_{27}$N$_4$S$_2$: $m/z$ 339.1672, $m/z$ found 339.1670.

## Synthesis of (5-bromo-1,3-phenylene)bis(methylene) dicarbamimidothioate, HBr salt (4 - Br-BIT)

Compound **4** was synthesized following method 1 using 1-Bromo-3,5-bis(bromomethyl) benzene (200 mg, 0.58 mmol, one eq) and thiourea (98 mg, 1.3 mmol, 2.2 eq) in ethanol (10 ml). The precipitate was washed with EtOAc and dried *in vacuo* to afford **4** (262 mg, 0.53 mmol, 90%) as a white solid (**Appendix 1—figure 4**).

RP-UPLC: $t_R$ = 1.25 min (method A). $^1$H NMR (400 MHz, DMSO): δ 9.13 (s, 6H), 7.63 (d, $J$ = 1.2 Hz, 2H), 7.46 (s, 1H), 4.50 (s, 4H). $^{13}$C NMR (101 MHz, DMSO): δ 168.5, 138.5, 131.2 (CH$_{ar}$), 128.5 (CH$_{ar}$), 121.8, 33.2 (CH$_2$). HR-MS calculated for C$_{10}$H$_{14}$BrN$_4$S$_2$: $m/z$ 332.9838, $m/z$ found 332.9836.

## Synthesis of (2-bromo-1,3-phenylene)bis(methylene) dicarbamimidothioate, TFA salt (5 - oBr-BIT)

Compound **5** was synthesized following method 1 using 2-bromo-1,3-bis(bromomethyl)-benzene (158 mg, 0.46 mmol, one eq) and thiourea (79 mg, 1.05 mmol, 2.3 eq) in ethanol (8 ml). The residue was purified by preparative HPLC (gradient from 100/0 A/D to 85/15 A/D in 18 min, $t_R$ = 4% D) to afford **5** (70 mg, 0.13 mmol, 27%) as a white solid (**Appendix 1—figure 5**).

RP-UPLC: $t_R$ = 1.18 min (method A). $^1$H NMR (400 MHz, DMSO): δ 9.36 (6H, s, NH + NH$_2$), 7.59 (2H, d, $J$ = 7.6 Hz, H$_{ar}$), 7.47 (1H, t, $J$ = 7.2 Hz, H$_{ar}$), 4.60 (4H, s, CH$_2$). $^{13}$C NMR (101 MHz, DMSO): δ 168.9, 135.5, 131.3 (CH$_{ar}$), 128.5 (CH$_{ar}$), 126.5, 35.8 (CH$_2$). HR-MS calculated for C$_{10}$H$_{14}$BrN$_4$S$_2$: $m/z$ 332.9838, $m/z$ found 332.9842.

## Synthesis of 2-bromo-3-(((4,5-dihydro-1H-imidazol-2-yl) thio)methyl)benzyl carbamimidothioate, TFA salt (compound 6)

2-bromo-1,3-bis(bromomethyl)-benzene (150 mg, 0.44 mmol, one eq) was dissolved in ethanol (8 mL) before the addition of thiourea (34 mg, 0.45 mmol, one eq). The solution was stirred at 40°C overnight and then cooled down to room temperature. Ethanol was removed and the residue was purified by column chromatography on silica gel (DCM/MeOH 90/10). The product was dissolved in ethanol (8 ml) and imidazolidine-2-thione (47 mg, 0.44 mmol, one eq) was added. The solution was stirred at 40°C overnight. The solution was cooled down to room temperature and ethanol was removed except 1 ml which was kept. Hexane was added and the precipitate was filtered and dried *in vacuo*. The product was purified by preparative HPLC (gradient from 100% A to 90/10 A/D in 28 min, $t_R$ = 7% D) to afford **6** (73 mg, 0.13 mmol, 28%) as a white solid (**Appendix 1—figure 6**).

RP-UPLC: $t_R$ = 1.21 min (method A). $^1$H NMR (300 MHz, MeOD): δ 7.58 (2H, d, $J$ = 7.6 Hz, H$_{ar}$), 7.47 (1H, t, $J$ = 7.5 Hz, H$_{ar}$), 4.67 (2H, s, CH$_2$), 4.63 (2H, s, CH$_2$), 4.00 (4H, s, CH$_2$). $^{13}$C NMR (101 MHz, MeOD): δ 172.0, 170.9, 136.6, 136.2, 133.1 (CH$_{ar}$), 132.9 (CH$_{ar}$), 129.9

(CH$_{ar}$), 128.0, 6, 46.8 (CH$_2$), 37.9 (CH$_2$), 37.8 (CH$_2$). HR-MS calculated for C$_{12}$H$_{16}$BrN$_4$S$_2$: $m/z$ 358.9994, $m/z$ found 358.9995.

## Synthesis of 2,2'-(((2-bromo-1,3-phenylene)bis (methylene))bis(sulfanediyl))bis(4,5-dihydro-1H-imidazole), TFA salt (compound 7)

Compound **7** was synthetized following method 1 using 2-bromo-1,3-bis(bromomethyl)-benzene (165 mg, 0.48 mmol, one eq) and imidazolidine-2-thione (111 mg, 1.1 mmol, 2.2 eq) in ethanol (8 ml). The residue was purified by preparative HPLC (gradient from 100% A to 80/20 A/D in 28 min, t$_R$ = 8% D) to afford **7** (141 mg, 0.23 mmol, 48%) as a white solid (*Appendix 1—figure 7*).

RP-UPLC: t$_R$ = 1.17 min (method A). $^1$H NMR (300 MHz, DMSO): δ 7.60 (2H, d, $J$ = 7.5 Hz, H$_{ar}$), 7.48 (1H, t, $J$ = 7.3 Hz, H$_{ar}$), 4.67 (4H, s, CH$_2$), 4.00 (8H, s, CH$_2$). $^{13}$C NMR (101 MHz, MeOD): δ 170.9, 136.3, 133.0 (CH$_{ar}$), 130.0 (CH$_{ar}$), 128.0, 46.8 (CH$_2$), 37.9 (CH$_2$). HR-MS calculated for C$_{14}$H$_{18}$BrN$_4$S$_2$: $m/z$ 385.0151, $m/z$ found 385.0142.

## Synthesis of 4,6-dimethyldibenzo[b,d]furan (8)

In a three-necks flask, dibenzofuran (2.0 g, 11.9 mmol, one eq) was dissolved in dry diethyl ether (100 ml) under argon atmosphere. The temperature was cooled down to – 78°C before the addition of TMEDA (4.3 ml, 29.5 mmol, 2.6 eq). *sec* BuLi (21.2 ml, 1.4 M, 29.7 mmol, 2.5 eq) was added slowly (dropwise). The mixture was stirred 16 hr and allowed to heat at room temperature during the night. Iodomethane (3.80 ml, 61.0 mmol, 5.0 eq) was added. The reaction mixture was stirred at room temperature for 20 hr. Then NH$_4$Cl *sat* (70 ml) was added to quench the reaction. The product was extracted with diethyl ether (3 times 100 ml). The organic layers were combined, dried over Na$_2$SO$_4$, filtered and concentrated. The residue was recrystallized from methanol to afford **8** (846 mg, 4.3 mmol, 36%) as a white solid (*Chafeev et al., 2008*).

$^1$H NMR (400 MHz, CDCl$_3$): δ 7.75 (2H, dd, $J$ = 6.4, 2.6 Hz, H$_{ar}$), 7.25–7.20 (2H, m, H$_{ar}$), 2.62 (6H, s, CH$_3$). $^{13}$C NMR (75 MHz, CDCl$_3$): δ 155.1, 128.0 (CH$_{ar}$), 124.2, 122.6 (CH$_{ar}$), 122.0, 118.2 (CH$_{ar}$), 15.4 (CH$_3$).

## Synthesis of 3,7-dibromo-4,6-dimethyldibenzo[b,d]furan (9)

**8** (200 mg, 1.0 mmol, one eq) was dissolved in acetic acid (3 ml). Br$_2$ (0.11 ml, 2.1 mmol, 2.1 eq) was added. The mixture was stirred at room temperature overnight. Br$_2$ (0.4 ml, 0.82 mmol, 0.8 eq) was added and the solution was stirred at room temperature for 24 hr. Water was added. A precipitate appeared, was filtered and kept. DCM was added in the filtrate. Product was extracted with DCM. Organic layers were combined, washed with water, dried over Na$_2$SO$_4$ and concentrated. All compound was recrystallized from ethyl acetate and then purified by flash column chromatography on silica gel (Hexane) to afford **9** (330 mg, 1.1 mmol, 65%) as an off-white solid (*Chafeev et al., 2008*).

RP-GCMS: t$_R$ = 9.50 min (from 50°C to 280°C in 8 min, then stay at 280°C). $^1$H NMR (300 MHz, CDCl$_3$): δ 7.58 (2H, d, $J$ = 8.4 Hz), 7.50 (2H, d, $J$ = 8.3 Hz), 2.64 (s, 6H). $^{13}$C NMR (101 MHz, CDCl$_3$): δ 155.3, 127.2 (CH$_{ar}$), 123.3, 123.0, 122.9, 118.8 (CH$_{ar}$), 15.7 (CH$_3$).

## Synthesis of 3,7-dibromo-4,6-bis(bromomethyl)dibenzo [b,d]furan (10)

**9** (196 mg, 0.56 mmol, 1.0 eq) and NBS (214 mg, 1.2 mmol, 2.2 eq) were dissolved in CCl$_4$ (41 ml) under argon. Benzoyl peroxide (one spatula) was added. The mixture was stirred at 60°C overnight. NBS (212 mg, 1.2 mmol, 2.2 eq) and benzoyl peroxide (1.5 spatula) were added. The mixture was stirred at 60°C overnight before cooled down to 0°C. The white precipitate was filtered, and the solvent was removed. The residue was recrystallized from

Hexane/Ethyl acetate (50:50) to afford **10** (134 mg, 0.56 mmol, 47%) as an off-white solid (*Chafeev et al., 2008*).

$^1$H NMR (300 MHz, CDCl$_3$): δ 7.73 (2H, d, *J* = 8.3 Hz), 7.60 (2H, d, *J* = 8.3 Hz), 5.00 (4H, s, CH$_2$). $^{13}$C NMR (101 MHz, CDCl$_3$): δ 154.9, 128.4 (CH$_{ar}$), 123.6, 123.6, 122.8, 121.8 (CH$_{ar}$), 26.0 (CH$_2$).

## Synthesis of 2,4-bis(bromomethyl)−1,3,5-trimethylbenzene (11)

To a suspension of paraformaldehyde (150 mg, 5.0 mmol, two eq) in acetic acid (2.5 ml), mesitylene (0.35 ml, 2.5 mmol, one eq) was added. HBr (1 ml, 31% in acetic acid) was added quickly. The mixture was stirred at 80°C for 8 hr. Water (10 ml) was added and the white precipitate was filtered and dried *in vacuo*. The product was purified by flash chromatography on silica gel (Hexane) to afford **11** (530 mg, 1.7 mmol, 69%) as a white solid (*Van der Made and Van der Made, 1993*).

$^1$H NMR (400 MHz, CDCl$_3$): δ 6.90 (s, 1H), 4.57 (s, 4H), 2.45 (s, 3H), 2.38 (s, 6H). $^{13}$C NMR (101 MHz, CDCl$_3$): δ 138.2, 137.3, 132.8, 130.9 (CH$_{ar}$), 29.9 (CH$_2$), 19.6 (CH$_3$), 14.8 (CH$_3$).

## Synthesis of 2,4-bis(bromomethyl)−1,3,5-triethylbenzene (12)

To a suspension of paraformaldehyde (80 mg, 2.7 mmol, two eq) in acetic acid (2.5 ml), 1,3,5-triethylbenzene (0.25 ml, 1.3 mmol, one eq) was added. HBr (0.7 ml, 31% in acetic acid) was added quickly. The mixture was stirred at 80°C for 8 hr. Water (10 ml) was added and product was extracted three times with EtOAc. The combined organic layers were dried over Na$_2$SO$_4$ and concentrated. The residue was purified by flash chromatography on silica gel (Hexane) to afford **12** (42 mg, 0.1 mmol, 7%) as a white solid (*Van der Made and Van der Made, 1993*).

RP-UPLC: $t_R$ = 1.91 min (method A). $^1$H NMR (400 MHz, CDCl$_3$): δ 6.97 (1H, s, H$_{ar}$), 4.60 (4H, s, CH$_2$), 2.94 (2H, q, *J* = 7.6 Hz, C*H$_2$*CH$_3$), 2.77 (4H, q, *J* = 7.6 Hz, C*H$_2$*CH$_3$), 1.32 (9H, m, CH$_3$). $^{13}$C NMR (101 MHz, CDCl$_3$): δ 144.9, 143.6, 131.5, 127.7 (CH$_{ar}$), 28.7 (CH$_2$), 26.0 (CH$_2$), 22.5 (CH$_2$), 15.9 (CH$_3$), 15.1 (CH$_3$).

**Appendix 1—table 2.** HPLC Purity of final compounds.

| Compound | Retention time | Purity [%] | HPLC method[*] |
|---|---|---|---|
| 1 | 3.27 min | 99 | B |
| 2 | 1.31 min | 99 | A |
| 3 | 1.52 min | 99 | A |
| 4 | 1.25 min | 99 | A |
| 5 | 1.18 min | 99 | A |
| 6 | 1.21 min | 99 | A |
| 7 | 1.17 min | 99 | A |

[*]Method A: in 2.2 min from 100% A to 100% D, then staying on 100% D. Method B: in 7.5 min from 100% A to 100% D, then staying on 100% D.

