## [Decision Letter]

**Acceptance summary:**

The human divalent metal ion transporter (DMT-1) plays a major role in the absorption of Fe2+ iron in the intestine. Its dysregulation can cause iron overload disorder such as hereditary hemochromatosis, for which small molecule inhibitors are being developed as potential therapeutics. One class of such inhibitors are TMBIT and TEBIT. The manuscript by Manatschal and colleagues describes their work on the characterization of these compounds with human DMT-1 as well as its bacterial homolog EcoDMT. Using a combination of X-ray crystallography, binding assays and transport assays in whole cells and reconstituted proteoliposomes, they were able to show that a bromine derivative Br-BIT binds to the inner end of the funnel in the outward-facing conformation in EcoBIT and that the drug binding site shares residues with the metal ion site, agreeing with the competitive inhibition observed in the biochemistry data. The authors further demonstrated that the drugs bind and inhibit the human transporter in a similar manner, and confirmed the drug binding sites using mutagenesis. This study is a beautiful example of how structural and functional studies can be combined to provide atomic level insight into drug mechanisms. The results will be useful for the development of drugs targeting hereditary hemochromatosis patients.

**Decision letter after peer review:**

Please address the following comments in your final submission:

1) Introduction: APC – please define

2) Subsection “Functional characterization of the interaction of bis-isothiurea substituted aromatic 106 compounds with human DMT1”: with BOTH isothiourea groups being predominantly charged

3) Subsection “Functional characterization of effects of mutations… on inhibition”

4) Figure 1—figure supplement 1A: Please specify the volume of titrant (TMBIT, Br-BIT), or convert abscissa unit to mol/mol, to facilitate interpretation of the tiration curves (both IT groups are presumably protonated at neutral pH). You might also consider merging the two panels, with HCl titration depicted on the negative side of the abscissa.

5) Figure 2B: "The solid lines are fits to the Michaelis-Menten equation". In principle one would expect a double M-M curve (50% of transporters with normal Km, 50% with increased Km (Km')). Even if the difference between Km' and Km might be too small to resolve those two populations, this fact should be at least mentioned for clarity.

6) Table 4: Please provide units for IC50 (uM). Also, since for a competitive mechanism IC50 depends on <S>, please provide that information as well. (E.g., the IC50 values for WT in Table 4 differ from those listed in Figure 1—figure supplement 2 in which 1 μm Fe2+ was used as substrate.)

7) A 2D contact diagram should be included to show the interactions between the drug with EcoDMT and between EcoDMT with the metal ion.

8) The affinity of the current compounds to EcoCMT and DMT-1 is still relatively low. Consider discussing how this work might suggest new ways to modify the small molecules to increase the affinity.

9) The last sentence of paragraph 2 of the Introduction is very long. It might be better to split it into two sentences.

---

## [Author Response]

Please address the following comments in your final submission:1) Introduction: APC – please define

We have defined APC (amino acid-polyamine-cation) in the manuscript.

2) Subsection “Functional characterization of the interaction of bis-isothiurea substituted aromatic 106 compounds with human DMT1”: with BOTH isothiourea groups being predominantly charged

We have introduced the correction.

3) Subsection “Functional characterization of effects of mutations.… on inhibition”

We have changed the subsection to: “Functional characterization of inhibitor binding-site mutants of EcoDMT” to be consistent with the following chapter.

4) Figure 1—figure supplement 1A: Please specify the volume of titrant (TMBIT, Br-BIT), or convert abscissa unit to mol/mol, to facilitate interpretation of the tiration curves (both IT groups are presumably protonated at neutral pH). You might also consider merging the two panels, with HCl titration depicted on the negative side of the abscissa.

We have merged the two panels and changed the units to equivalents of the titrant (OH- or H^+^).

5) Figure 2B: "The solid lines are fits to the Michaelis-Menten equation". In principle one would expect a double M-M curve (50% of transporters with normal Km, 50% with increased Km (Km')). Even if the difference between Km' and Km might be too small to resolve those two populations, this fact should be at least mentioned for clarity.

A unique fit with two Km values is limited by the data, we have thus added the following statement:

“The solid lines are fits to the Michaelis-Menten equation assuming similar kinetic properties of transport for both orientations of the transporter. The observed kinetic parameters thus describe apparent values obtained from an average of transport properties in inside-out and outside-out orientations.”

6) Table 4: Please provide units for IC50 (uM). Also, since for a competitive mechanism IC50 depends on <S>, please provide that information as well. (E.g., the IC50 values for WT in Table 4 differ from those listed in Figure 1—figure supplement 2 in which 1 μm Fe2+ was used as substrate.)

We have provided the units in the title and added the following sentence to the footnotes:

The values shown for WT deviate from the values shown in Figure 1—figure supplement 2, due to small differences in the experimental setup (*i.e.* the use of a stable cell line *vs.* transiently transfected cells).

7) A 2D contact diagram should be included to show the interactions between the drug with EcoDMT and between EcoDMT with the metal ion.

We have introduced a schematic 2D figure showing the interactions of the inhibitor and the metal ion with its binding site as panel B in the revised Figure 3—figure supplement 1.

8) The affinity of the current compounds to EcoCMT and DMT-1 is still relatively low. Consider discussing how this work might suggest new ways to modify the small molecules to increase the affinity.

We have introduced the following sentence in the Discussion section:

“A strategy to increase the potency and selectivity of compounds towards human DMT1 could thus rely on the optimization of interactions at the distal side of the binding pocket by a systematic variation of aromatic scaffolds and attached polar groups.”

9) The last sentence of paragraph 2 of the Introduction is very long. It might be better to split it into two sentences.

We have split the sentence:

“The transport properties of DMT1 have been characterized in detail by employing electrophysiology and cellular uptake studies, which demonstrated the broad selectivity for transition metals and the discrimination of alkaline earth metal ions such as calcium. The same studies also revealed the symport of H^+^, which serves as energy source for concentrating the substrate in the cell.”